# Target Localization and Sensor Movement Trajectory Planning with Bearing-Only Measurements in Three Dimensional Space

**Yiqun Zou** [1], **Bilu Gao** [1], **Xiafei Tang** [2] and **Lingli Yu** [1,*]

1   School of Automation, Central South University, Changsha 410083, China; yiqunzou@csu.edu.cn (Y.Z.); bilugao@csu.edu.cn (B.G.)
2   School of Electrical and Information Engineering, Changsha University of Science and Technology, Changsha 410114, China; xiafei.tang@csust.edu.cn
*   Correspondence: llyu@csu.edu.cn

**Abstract:** In order to improve the accuracy of bearing-only localization in three dimensional (3D) space, this paper proposes a novel bias compensation method and a new single-sensor maneuvering trajectory algorithm, respectively. Compared with traditional methods, the bias compensation method estimates the unknown variance of bearing noise consistently, which is utilized in pseudo-linear target localization to achieve higher precision. The sensor maneuvering algorithm designs the next moment sensor location in consideration of all the past sensor locations, unlike other approaches that only consider finite past locations. Research shows that the trajectories generated by our algorithm have greater Fisher information matrix (FIM) determinants and better localization accuracy.

**Keywords:** bearing-only localization; three dimensional space; bias compensation method; Fisher information matrix (FIM) determinant; sensor maneuvering algorithm

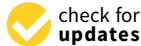

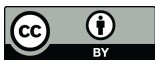

## 1. Introduction

Target localization has applications in both civilian and military domains including wireless communications, environmental monitoring [1], network security [2,3], medical diagnosis, etc. The commonly used methods for target localization can be characterized into four categories according to the type of sensors: bearing-only (also called angle of arrival (AOA)), range-only, time-of-arrival (time-difference-of-arrival) and received signal strength [4]. The problem of using only angle measurements to locate a target is referred to as bearing-only target localization which is the focus of this paper.

The bearing-only target positioning problem can usually be solved in the following two categories. The first category is based on Kalman filtering and its various evolutionary versions [5,6], Kalman filter, extended Kalman filter and unscented Kalman filter have been applied to the bearing-only target localization problem in [7,8]. Due to nonlinear-to-linear transformation of the angle measurement equations, there exists a serious deviation problem. Dogançay et al. [9] propose a new two-step pseudo-linear Kalman filter for bearing-only target tracking in 3D space. In the second category, many batch form methods based on the ordinary least squares are used to deal with target localization under small measurement noise [10–12]. To deal with the serious deviation of the pseudo-linear estimator (PLE) in two dimensional (2D) target localization problem, Dogançay [13] proposes a closed-form reduced-bias PLE to realize the asymptotic unbiased estimation of target motion parameters by using the instrumental variable (IV) method. In the case of small noise, Dogançay also proposes pseudo-linear estimator for AOA target motion analysis in 3D space [14]. Adib et al. [15] proposes a two-stage weighted Stansfield geolocation algorithm based on maximum likelihood estimation in 3D space, and it has the advantages of consistency and statistical efficiency. In addition, the position of the sensor has a great influence on positioning accuracy [4,16]. In order to minimize the uncertainty of the target estimation, the position of the sensor at the next moment needs to be reasonably planned. The optimal localization of the sensor is generally determined by maximizing

the determinant of FIM or minimizing the trace of error covariance matrix of the estimator [2,17–19]. The optimal trajectory of a single mobile sensor is determined by maximizing the determinant of FIM under the condition of adding state constraints on the observer trajectory [20]. When the absolute elevation of the sensor is constant, Sheng et al. [21] proposes a new simple sensor optimal deployment criterion based on minimizing the trace of error covariance matrix. Based on the optimization criterion of minimizing the mean square error (MSE), an own-ship path optimization algorithm by using the gradient-descent method in $xy$ plane and the grid search optimization method in $z$ axis is presented [22]. A distributed path optimization algorithm is applied when the communication distance constraints and no-fly zone are considered [23].

Most estimation methods generally assume that the variance of measurement noise is known a priori for the purpose of bias compensation. In fact, the noise variance is generally unknown and needs to be approximated in advance. The least-square solution for the target dynamics is calculated and plugged into bearing measurement equations to compute the noise in the first step. Then the "true" noise is used to construct the pseudolinear vector and form the final bias-compensation PLE result which is still biased [24]. As the first contribution of this paper, we analyze the properties of Gaussian noise along with the angle measurement equations and newly discover the potential relationship between the determinant of the extended coefficient matrix of the equation and the noise variance. A BC method for target localization on $xy$ plane and the target position in $z$-axis direction based on the pseudo-linear estimation method is proposed thereafter.

Optimizing the increment of FIM determinant over a finite time horizon rather than the whole time span makes trajectory planning less effective [25]. As the second contribution of this paper, a sensor trajectory planning algorithm based on greedy strategies maximizes the increment between two consecutive FIM determinants to realize the next location of a single sensor. It is proven that the increment equals the aggregation of FIM determinants constituted by the sensor location at next time step and the combination of all past locations. Considering the spherical constraints of target safety area and sensor movement area, the optimal solution always sits on the area surface. Based on this fact, an analytical solution is suggested to solve the maximization problem.

The rest of this paper is organized as follows. Section 2 formulates the problem of target localization in 3D space, including the notations and localization methods. Section 3 proposes a BC estimator and bias compensation weighted instrumental variable (BC-WIV) estimator. In Section 4, the sensor maneuvering algorithm in the composition of two trajectories is designed in detail with the analysis of the FIM increment. Numerical examples are studies to validate the method studies in Section 5. Section 6 concludes the whole paper and points out future research directions.

## 2. Bearing-Only Localization in 3D Space

As shown in Figure 1, the location of target $\mathbf{T} = \begin{bmatrix} x_t & y_t & z_t \end{bmatrix}'$ is unknown in 3D space. $\mathbf{T}$ needs to be estimated based on the azimuth $\hat{\phi}_i$ and elevation angle $\hat{\theta}_i$ measurements collected from a mobile sensor $\mathbf{S}_i = \begin{bmatrix} x_{s_i} & y_{s_i} & z_{s_i} \end{bmatrix}'$ $(i = 1, \ldots, n)$ at $n$ locations. $\mathbf{t}$ and $\mathbf{s_i}$ are the coordinates of the target and the sensor on $xy$ plane, respectively. The true bearing information $\phi_i$ and $\theta_i$ received by the sensor from the target are:

$$\phi_i = \tan^{-1}\left(\frac{y_t - y_{s_i}}{x_t - x_{s_i}}\right), \tag{1}$$

$$\theta_i = \tan^{-1}\left(\frac{z_t - z_{s_i}}{\sqrt{(x_t - x_{s_i})^2 + (y_t - y_{s_i})^2}}\right). \tag{2}$$

The bearing information $\hat{\phi}_i$ and $\hat{\theta}_i$ are the sensor measurements corrupted by the noise $e_i$ and $\delta_i$ at n locations, i.e.,

$$\hat{\phi}_i = \phi_i + e_i, \tag{3}$$

$$\hat{\theta}_i = \theta_i + \delta_i, \tag{4}$$

where $e_i$ and $\delta_i$ are independent identically distributed white Gaussian noise with zero mean and unknown variance of $\sigma^2$ and $\xi^2$, respectively.

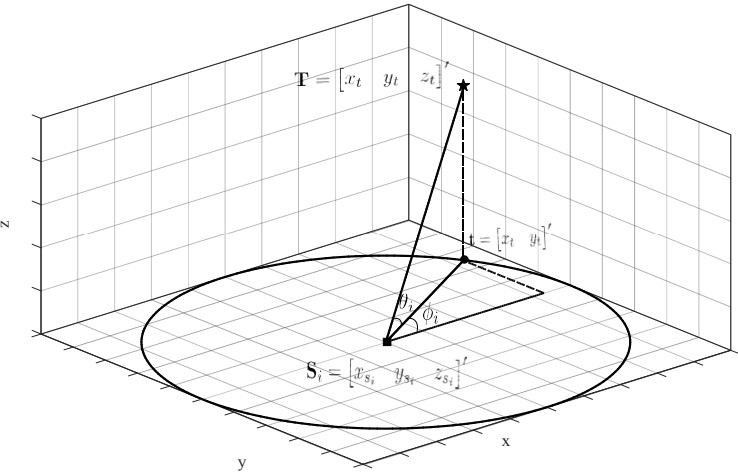

**Figure 1.** Target and sensor in $\mathbb{R}^3$.

The problem of target localization in 3D space is solved in two steps. Firstly, the bearing line is projected onto the $xy$ plane. The corresponding 2D estimator is used to solve the resulting 2D positioning problem. Then the estimated position of **T** along the $z$ axis is obtained from the estimated $x$ and $y$ coordinates. The 2D pseudolinear estimator (PLE) algorithm is used to estimate **t** by using $n$ azimuth measurements $\hat{\phi}_i$. To turn a nonlinear problem into a linear one, small noise assumption also appears in the development and analysis of the Stansfield algorithm [26] and the total-least-squares solution [27] for localization. The pseudolinear equation from (1) and (3) between **t** and the given information is described as:

$$\sin \hat{\phi}_i x_t - \cos \hat{\phi}_i y_t = \sin \hat{\phi}_i x_{s_i} - \cos \hat{\phi}_i y_{s_i} + ||\mathbf{t} - \mathbf{s}_i|| \sin e_i, \tag{5}$$

which can be expressed in matrix form as:

$$\mathbf{A}_n \mathbf{t} = \mathbf{B}_n + \mathbf{n}_n \tag{6}$$

where

$$\mathbf{A}_n = \begin{bmatrix} \sin \hat{\phi}_1 & -\cos \hat{\phi}_1 \\ \vdots & \vdots \\ \sin \hat{\phi}_n & -\cos \hat{\phi}_n \end{bmatrix}, \mathbf{B}_n = \begin{bmatrix} \sin \hat{\phi}_1 x_{s_1} - \cos \hat{\phi}_1 y_{s_1} \\ \vdots \\ \sin \hat{\phi}_n x_{s_n} - \cos \hat{\phi}_n y_{s_n} \end{bmatrix},$$

$$\mathbf{n}_n = \begin{bmatrix} ||\mathbf{t} - \mathbf{s}_1|| & 0 & 0 \\ 0 & \ddots & 0 \\ 0 & 0 & ||\mathbf{t} - \mathbf{s}_n|| \end{bmatrix} \begin{bmatrix} \sin e_1 \\ \vdots \\ \sin e_n \end{bmatrix}. \tag{7}$$

The 2D PLE estimator of (6) is:

$$\hat{\mathbf{t}}_{\mathbf{ple}} = (\mathbf{A}'_n \mathbf{A}_n)^{-1} \mathbf{A}'_n \mathbf{B}_n, \tag{8}$$

where $\mathbf{A}'_n$ represents the transpose of $\mathbf{A}_n$.

The target position of the projection in 2D space has been estimated. According to the geometric relationship, the target position in the $z$ axis direction can be estimated according to the elevation angle measurements and other information as:

$$z_0 = \frac{1}{n} \sum_{i=1}^{n} \left( z_i + \sqrt{(x_0 - x_{s_i})^2 + (y_0 - y_{s_i})^2} \tan \hat{\theta}_i \right),\tag{9}$$

where $x_0$ and $y_0$ are the target position estimated by different methods in 2D space.

## 3. Bias Compensation Estimator for Localization in 3D Space

### 3.1. Bias Compensation Localization in 2D Space

In [28], a bias compensation method for bearing-only localization is proposed, which can effectively solve the problem of nonlinear localization. Dividing $\cos \hat{\phi}_i$ on both sides of (5) gives:

$$\tan \hat{\phi}_i x_t - y_t = \tan \hat{\phi}_i x_{s_i} - y_{s_i} + \frac{||\mathbf{t} - \mathbf{s}_i|| \sin e_i}{\cos \hat{\phi}_i}.\tag{10}$$

Rewrite (10) into the matrix form as:

$$\mathbf{Z}_n = \bar{\mathbf{W}}_n \mathbf{t} + \mathbf{C}_n \mathbf{N}_n,\tag{11}$$

where

$$\mathbf{Z}_n = \begin{bmatrix} \tan \hat{\phi}_1 x_{s_1} - y_{s_1} \\ \vdots \\ \tan \hat{\phi}_n x_{s_n} - y_{s_n} \end{bmatrix}, \bar{\mathbf{W}}_n = \begin{bmatrix} \tan \hat{\phi}_1 & -1 \\ \vdots & \vdots \\ \tan \hat{\phi}_n & -1 \end{bmatrix},$$

$$\mathbf{C}_n = \begin{bmatrix} -\dfrac{||\mathbf{t} - \mathbf{s_1}||}{\cos \hat{\phi}_1} & 0 & 0 \\ 0 & \ddots & 0 \\ 0 & 0 & -\dfrac{||\mathbf{t} - \mathbf{s_n}||}{\cos \hat{\phi}_n} \end{bmatrix}, \mathbf{N}_n = \begin{bmatrix} \sin e_1 \\ \vdots \\ \sin e_n \end{bmatrix}.\tag{12}$$

Based on

$$\hat{\sigma}^2 = \min \lambda,$$

$$s.t. \ \det \left( \frac{1}{n} \begin{bmatrix} \bar{\mathbf{W}}_n & \mathbf{Z}_n \end{bmatrix}' \begin{bmatrix} \bar{\mathbf{W}}_n & \mathbf{Z}_n \end{bmatrix} - \lambda \begin{bmatrix} 1 & 0 & -\frac{1}{n}\sum_{i=1}^{n} x_{s_i} \\ 0 & 0 & 0 \\ -\frac{1}{n}\sum_{i=1}^{n} x_{s_i} & 0 & \frac{1}{n}\sum_{i=1}^{n} x_{s_i}^2 \end{bmatrix} \right) = 0\tag{13}$$

of $\sigma^2$, the Bai estimator is:

$$\hat{t}_{\hat{\sigma}^2} = \left( \frac{1}{n}\bar{\mathbf{W}}_n'\bar{\mathbf{W}}_n - \hat{\sigma}^2 \begin{bmatrix} 1 & 0 \\ 0 & 0 \end{bmatrix} \right)^{-1} \left( \frac{1}{n}\bar{\mathbf{W}}_n'\mathbf{Z}_n + \hat{\sigma}^2 \begin{bmatrix} \frac{1}{n}\sum_{i=1}^{n} x_{s_i} \\ 0 \end{bmatrix} \right).\tag{14}$$

Inspired by [28], a corresponding bias compensation estimator is designed for (5). Firstly, the matrix $\mathbf{A}_n$ and $\mathbf{B}_n$ are processed as:

$$\frac{1}{n} \begin{bmatrix} \mathbf{A}_n & \mathbf{B}_n \end{bmatrix}' \begin{bmatrix} \mathbf{A}_n & \mathbf{B}_n \end{bmatrix} = \frac{1}{n} \sum_{i=1}^{n} \begin{bmatrix} a_{11}^i & a_{12}^i & b_1^i \\ a_{12}^i & a_{22}^i & b_2^i \\ b_1^i & b_2^i & c^i \end{bmatrix},\tag{15}$$

where

$$a_{11}^i = \sin^2 \phi_i \cos^2 e_i + 2 \sin \phi_i \cos \phi_i \cos e_i \sin e_i + \cos^2 \phi_i \sin^2 e_i, \tag{16}$$

$$a_{12}^i = -\sin \phi_i \cos \phi_i \cos^2 e_i - \cos^2 \phi_i \cos e_i \sin e_i$$
$$+ \sin^2 \phi_i \cos e_i \sin e_i + \cos \phi_i \sin \phi_i \sin^2 e_i, \tag{17}$$

$$a_{22}^i = \cos^2 \phi_i \cos^2 e_i - 2 \cos \phi_i \sin \phi_i \cos e_i \sin e_i + \sin^2 \phi_i \sin^2 e_i, \tag{18}$$

$$b_1^i = a_{11}^i x_{s_i} + a_{12}^i y_{s_i}, \tag{19}$$

$$b_2^i = a_{12}^i x_{s_i} + a_{22}^i y_{s_i}, \tag{20}$$

$$c^i = a_{11}^i x_{s_i}^2 + 2 a_{12}^i x_{s_i} y_{s_i} + a_{22}^i y_{s_i}^2. \tag{21}$$

To derive the bias compensation estimator in detail, two lemmas are proposed.

**Lemma 1.** $\mathbb{E}(\cos e_i \sin e_i) = 0$, $\mathbb{E}(\sin^2 e_i) = (1 - e^{-2\sigma^2})/2$, $\mathbb{E}(\cos^2 e_i) = (1 + e^{-2\sigma^2})/2$ where $\mathbb{E}(\cdot)$ represents the mathematical expectation.

**Proof of Lemma 1.** Since $\cos e_i \sin e_i = \sin 2e_i/2$ is an odd function, we have:

$$\frac{1}{2\sqrt{2\pi}\sigma} \int_{-\infty}^{\infty} \sin 2e_i \cdot e^{-\frac{e_i^2}{2\sigma^2}} de_i = 0. \tag{22}$$

From

$$\begin{aligned}
\mathbb{E}(\cos^2 e_i - \sin^2 e_i) &= \mathbb{E}(\cos 2e_i) \\
&= \sum_{i=0}^{\infty} \frac{(-1)^i 2^{2i}}{(2i)!} \cdot \frac{1}{\sqrt{2\pi}\sigma} \int_{-\infty}^{\infty} e_i^{2i} e^{-\frac{e_i^2}{2\sigma^2}} de_i \\
&= \sum_{i=0}^{\infty} \frac{(-1)^i 2^{2i}}{(2i)!} \cdot \frac{\sigma^{2i}(2i)!}{2^i(i)!} \\
&= \sum_{i=0}^{\infty} \frac{(-1)^i (2\sigma^2)^i}{(i)!} \\
&= e^{-2\sigma^2}
\end{aligned} \tag{23}$$

and $\mathbb{E}(\cos^2 e_i + \sin^2 e_i) = 1$, $\mathbb{E}(\sin^2 e_i) = (1 - e^{-2\sigma^2})/2$, $\mathbb{E}(\cos^2 e_i) = (1 + e^{-2\sigma^2})/2$. $\square$

Define

$$P(\gamma) = \frac{1}{n} \begin{bmatrix} \mathbf{A}_n & \mathbf{B}_n \end{bmatrix}' \begin{bmatrix} \mathbf{A}_n & \mathbf{B}_n \end{bmatrix} - \frac{\gamma}{n} \begin{bmatrix} n & 0 & \sum\limits_{i=1}^{n} x_{s_i} \\ 0 & n & \sum\limits_{i=1}^{n} y_{s_i} \\ \sum\limits_{i=1}^{n} x_{s_i} & \sum\limits_{i=1}^{n} y_{s_i} & \sum\limits_{i=1}^{n} x_{s_i}^2 + \sum\limits_{i=1}^{n} y_{s_i}^2 \end{bmatrix} \tag{24}$$

as the extended coefficient matrix.

**Lemma 2.** $\gamma_{\min} = \mathbb{E}(\sin^2 e_i)$ is the smallest real value s.t. $\det(P(\gamma)) = 0$ in probability as $n \to \infty$.

**Proof of Lemma 2.** With Lemma 1, we have:

$$\frac{1}{n}\begin{bmatrix}\mathbf{A}_n & \mathbf{B}_n\end{bmatrix}'\begin{bmatrix}\mathbf{A}_n & \mathbf{B}_n\end{bmatrix} - P(\gamma_{\min}) = \frac{\gamma_{\min}}{n}\begin{bmatrix} n & 0 & \sum_{i=1}^{n}x_{s_i} \\ 0 & n & \sum_{i=1}^{n}y_{s_i} \\ \sum_{i=1}^{n}x_{s_i} & \sum_{i=1}^{n}y_{s_i} & \sum_{i=1}^{n}x_{s_i}^2 + \sum_{i=1}^{n}y_{s_i}^2 \end{bmatrix}$$

in probability where:

$$P(\gamma_{\min}) = \frac{\mathbb{E}(\cos(2e_i))}{n} * \sum_{i=1}^{n}\left(\begin{bmatrix}\sin\phi_i \\ -\cos\phi_i \\ \sin\phi_i x_{s_i} - \cos\phi_i y_{s_i}\end{bmatrix}\begin{bmatrix}\sin\phi_i \\ -\cos\phi_i \\ \sin\phi_i x_{s_i} - \cos\phi_i y_{s_i}\end{bmatrix}'\right). \qquad (25)$$

Through observation, $P(\gamma_{\min})$ is positive semi-definite. Its null space is one dimensional in the form of: $\beta\begin{bmatrix}x_t & y_t & -1\end{bmatrix}'$, where $\beta \in \mathbb{R}$. Let vector $\mathbf{v} = \begin{bmatrix}v_1 & v_2 & v_3\end{bmatrix}$ be arbitrary in $\mathbb{R}^{3\times1}$. So we have

$$\mathbf{v}'P(\gamma)\mathbf{v} = \mathbf{v}'P(\gamma_{\min})\mathbf{v} - \frac{\gamma - \gamma_{\min}}{n} * \mathbf{v}'\begin{bmatrix} n & 0 & \sum_{i=1}^{n}x_{s_i} \\ 0 & n & \sum_{i=1}^{n}y_{s_i} \\ \sum_{i=1}^{n}x_{s_i} & \sum_{i=1}^{n}y_{s_i} & \sum_{i=1}^{n}x_{s_i}^2 + \sum_{i=1}^{n}y_{s_i}^2 \end{bmatrix}\mathbf{v}. \qquad (26)$$

The second term

$$\frac{\gamma - \gamma_{\min}}{n}\mathbf{v}'\begin{bmatrix} n & 0 & \sum_{i=1}^{n}x_{s_i} \\ 0 & n & \sum_{i=1}^{n}y_{s_i} \\ \sum_{i=1}^{n}x_{s_i} & \sum_{i=1}^{n}y_{s_i} & .\sum_{i=1}^{n}x_{s_i}^2 + \sum_{i=1}^{n}y_{s_i}^2 \end{bmatrix}\mathbf{v}$$

$$= \frac{\gamma - \gamma_{\min}}{n}\sum_{i=1}^{n}(v_1^2 + 2v_1v_3x_{s_i} + v_2^2 + 2v_2v_3y_{s_i} + v_3^2x_{s_i}^2 + v_3^2y_{s_i}^2) \qquad (27)$$

$$= (\gamma - \gamma_{\min})((v_1 + \frac{v_3}{n}\sum_{i=1}^{n}x_{s_i})^2 + (v_2 + \frac{v_3}{n}\sum_{i=1}^{n}y_{s_i})^2$$

$$+ \frac{v_3^2}{2n^2}\sum_{i=1}^{n}\sum_{j=1}^{n}(x_{s_i} - x_{s_j})^2 + \frac{v_3^2}{2n^2}\sum_{i=1}^{n}\sum_{j=1}^{n}(y_{s_i} - y_{s_j})^2) > 0,$$

when $\gamma < \gamma_{\min}$, the first term $\mathbf{v}'P(\gamma_{\min})\mathbf{v}$ in (26) is no less than 0. This makes $\mathbf{v}'P(\gamma)\mathbf{v}$ always greater than 0. Hence, the minimum $\gamma$ that leads to $\mathbf{v}'P(\gamma)\mathbf{v} = 0$ or $\det(P(\gamma)) = 0$ satisfies $\gamma = \gamma_{\min}$ when $\mathbf{v} = \beta\begin{bmatrix}x_t & y_t & -1\end{bmatrix}'$. $\square$

It is worthwhile to point out that $\gamma_{\min}$ can be found by getting the minimum root of the third-order polynomial function $\det(P(\gamma)) = 0$. To sum up, the 2D positioning results with bias compensation can be obtained, as

**Theorem 1.**

$$\hat{\mathbf{t}}_{\mathbf{bc}} = \left(\frac{1}{n}\mathbf{A}_n'\mathbf{A}_n - \begin{bmatrix}\gamma_{\min} & 0 \\ 0 & \gamma_{\min}\end{bmatrix}\right)^{-1} * \left(\frac{1}{n}\mathbf{A}_n'\mathbf{B}_n - \frac{\gamma_{\min}}{n}\sum_{i=1}^{n}\begin{bmatrix}x_{s_i} \\ y_{s_i}\end{bmatrix}\right) \qquad (28)$$

*converges to $\mathbf{t}$ in probability as $n \to \infty$ when $\mathbb{E}(\cos 2e_i) \neq 0$.*

**Proof of Theorem 1.** As $n \to \infty$,

$$\frac{1}{n}\mathbf{A}_n'\mathbf{A}_n - \begin{bmatrix} \gamma_{\min} & 0 \\ 0 & \gamma_{\min} \end{bmatrix} = \frac{\mathbb{E}(\cos(2e_i))}{n}\sum_{i=1}^n \left( \begin{bmatrix} \sin\phi_i \\ -\cos\phi_i \end{bmatrix} \begin{bmatrix} \sin\phi_i \\ -\cos\phi_i \end{bmatrix}' \right), \tag{29}$$

$$\frac{1}{n}\mathbf{A}_n'\mathbf{B}_n - \frac{\gamma_{\min}}{n}\sum_{i=1}^n \begin{bmatrix} x_{s_i} \\ y_{s_i} \end{bmatrix} = \frac{\mathbb{E}(\cos(2e_i))}{n}\sum_{i=1}^n \left( \begin{bmatrix} \sin\phi_i \\ -\cos\phi_i \end{bmatrix} \begin{bmatrix} \sin\phi_i \\ -\cos\phi_i \end{bmatrix}' \right) \begin{bmatrix} x_t \\ y_t \end{bmatrix}. \tag{30}$$

From (29) and (30), Theorem 1 stands when $\mathbb{E}(\cos 2e_i) \neq 0$. $\square$

*3.2. Bias Compensation Method in Z-Axis*

The geometric relationship in 3D space is:

$$z_t \cos\hat{\theta}_i = \hat{r}_i \sin\hat{\theta}_i + z_i \cos\hat{\theta}_i, \tag{31}$$

where $\hat{r}_i = \sqrt{\left(\hat{\mathbf{t}}_{\mathbf{bc}}(1) - x_{s_i}\right)^2 + \left(\hat{\mathbf{t}}_{\mathbf{bc}}(2) - y_{s_i}\right)^2}$. Rewrite (31) into the matrix form

$$\mathbf{Y}_n z_t = \mathbf{F}_n, \tag{32}$$

where

$$\mathbf{Y}_n = \begin{bmatrix} \cos\hat{\theta}_1 \\ \vdots \\ \cos\hat{\theta}_n \end{bmatrix}, \mathbf{F}_n = \begin{bmatrix} z_{s_1}\cos\hat{\theta}_1 + \hat{r}_1\sin\hat{\theta}_1 \\ \vdots \\ z_{s_n}\cos\hat{\theta}_n + \hat{r}_n\sin\hat{\theta}_n \end{bmatrix}. \tag{33}$$

Similarly, we handle the matrix $\mathbf{Y}_n$ and $\mathbf{F}_n$,

$$\frac{1}{n}\begin{bmatrix} \mathbf{Y}_n & \mathbf{F}_n \end{bmatrix}' \begin{bmatrix} \mathbf{Y}_n & \mathbf{F}_n \end{bmatrix} = \frac{1}{n}\sum_{i=1}^n \begin{bmatrix} d_1^i & d_2^i \\ d_2^i & d_3^i \end{bmatrix}, \tag{34}$$

where

$$d_1^i = \cos^2\theta_i\cos^2\delta_i - 2\cos\theta_i\sin\theta_i\cos\delta_i\sin\delta_i + \sin^2\theta_i\sin^2\delta_i, \tag{35}$$

$$\begin{aligned} d_2^i = z_{s_i}d_1^i + \hat{r}_i(\sin\theta_i\cos\theta_i\cos^2\delta_i - \sin^2\theta_i\cos\delta_i\sin\delta_i \\ + \cos^2\theta_i\cos\delta_i\sin\delta_i - \sin\theta_i\cos\theta_i\sin^2\delta_i), \end{aligned} \tag{36}$$

$$\begin{aligned} d_3^i = z_{s_i}^2 d_1^i + 2z_{s_i}\hat{r}_i(\sin\theta_i\cos\theta_i\cos^2\delta_i - \sin^2\theta_i\cos\delta_i\sin\delta_i + \cos^2\theta_i\cos\delta_i\sin\delta_i \\ - \sin\theta_i\cos\theta_i\sin^2\delta_i) + \hat{r}_i^2(\sin^2\theta_i\cos^2\delta_i + 2\sin\theta_i\cos\theta_i\sin\delta_i\cos\delta_i + \cos^2\theta_i\sin^2\delta_i). \end{aligned} \tag{37}$$

We also define

$$Q(\mu) = \frac{1}{n}\begin{bmatrix} \mathbf{Y}_n & \mathbf{F}_n \end{bmatrix}' \begin{bmatrix} \mathbf{Y}_n & \mathbf{F}_n \end{bmatrix} - \frac{\mu}{n}\begin{bmatrix} n & \sum_{i=1}^n z_{s_i} \\ \sum_{i=1}^n z_{s_i} & \sum_{i=1}^n z_{s_i}^2 + \sum_{i=1}^n \hat{r}_i^2 \end{bmatrix}. \tag{38}$$

According to Lemma 2, the same reasoning process can be concluded. $\mu_{\min} = \mathbb{E}(\sin^2\delta_i)$ is the smallest real value *s.t.* $\det(Q(\mu)) = 0$ in probability as $n \to \infty$. The difference is $\mu_{\min}$ can be found by getting the minimum root of a second-order equation $\det(Q(\mu)) = 0$.

**Theorem 2.**

$$\hat{z}_{bc} = \left(\frac{1}{n}\mathbf{Y}_n'\mathbf{Y}_n - \mu_{\min}\right)^{-1}\left(\frac{1}{n}\mathbf{Y}_n'\mathbf{F}_n - \frac{\mu_{\min}}{n}\sum_{i=1}^n z_{s_i}\right) \tag{39}$$

*converges to $z_t$ in probability as $n \to \infty$ when $\mathbb{E}(cos2\delta_i) \neq 0$.*

**Proof of Theorem 2.** As $n \to \infty$,

$$\frac{1}{n}\mathbf{Y}_n'\mathbf{Y}_n - \mu_{\min} = \frac{\mathbb{E}(\cos(2\delta_i))}{n}\sum_{i=1}^{n}\left(\cos\theta_i * \cos\theta_i\right), \tag{40}$$

$$\frac{1}{n}\mathbf{Y}_n'\mathbf{F}_n - \frac{\mu_{\min}}{n}\sum_{i=1}^{n} z_{s_i} = \frac{\mathbb{E}(\cos(2\delta_i))}{n}\sum_{i=1}^{n}\left(\cos\theta_i * \cos\theta_i * z_t\right). \tag{41}$$

From (40) and (41), Theorem 2 stands when $\mathbb{E}(\cos 2\delta_i) \neq 0$. $\square$

From (28) and (39), the 3D bias compensation estimator result is $\hat{\mathbf{T}}_{\mathbf{bc}} = \left[\hat{\mathbf{t}}_{\mathbf{bc}}; \hat{z}_{bc}\right]'$.

*3.3. BC-WIV Estimator in 3D Space*

To improve the accuracy, the researchers often apply the weighted instrumental variable algorithm by constructing instrumental variable matrix $\mathbf{G}_{\mathbf{wiv}}$ and weighted matrix $\mathbf{W}_{\mathbf{wiv}}$,

$$\mathbf{G}_{\mathbf{wiv}} = \begin{bmatrix} \sin\bar{\phi}_1 & -\cos\bar{\phi}_1 \\ \vdots & \vdots \\ \sin\bar{\phi}_n & -\cos\bar{\phi}_n \end{bmatrix}, \mathbf{W}_{\mathbf{wiv}} = \begin{bmatrix} ||\mathbf{t} - \mathbf{s}_1||^2 & 0 & 0 \\ 0 & \ddots & 0 \\ 0 & 0 & ||\mathbf{t} - \mathbf{s}_n||^2 \end{bmatrix}, \tag{42}$$

where $\bar{\phi}_i$ is calculated from 2D PLE estimator via (8) and

$$\bar{\phi}_i = \tan^{-1}\left(\frac{\hat{\mathbf{t}}_{\mathbf{ple}}(2) - y_{s_i}}{\hat{\mathbf{t}}_{\mathbf{ple}}(1) - x_{s_i}}\right). \tag{43}$$

Then the 2D weighted instrumental variable estimator can be expressed as:

$$\hat{\mathbf{t}}_{\mathbf{wiv}} = (\mathbf{G}_{\mathbf{wiv}}'\mathbf{W}_{\mathbf{wiv}}^{-1}\mathbf{A}_n)^{-1}\mathbf{G}_{\mathbf{wiv}}'\mathbf{W}_{\mathbf{wiv}}^{-1}\mathbf{B}_n. \tag{44}$$

Similarly, the accuracy of bias compensation method in 2D space can be improved by constructing matrix,

$$\mathbf{G}_{\mathbf{bc}} = \begin{bmatrix} \sin\phi_1' & -\cos\phi_1' \\ \vdots & \vdots \\ \sin\phi_n' & -\cos\phi_n' \end{bmatrix}, \mathbf{W}_{\mathbf{bc}} = \begin{bmatrix} ||\hat{\mathbf{t}}_{\mathbf{bc}} - \mathbf{s}_1||^2 & 0 & 0 \\ 0 & \ddots & 0 \\ 0 & 0 & ||\hat{\mathbf{t}}_{\mathbf{bc}} - \mathbf{s}_n||^2 \end{bmatrix}, \tag{45}$$

where

$$\phi_i' = \tan^{-1}\left(\frac{\hat{\mathbf{t}}_{\mathbf{bc}}(2) - y_{s_i}}{\hat{\mathbf{t}}_{\mathbf{bc}}(1) - x_{s_i}}\right). \tag{46}$$

The 2D bias compensation weighted instrumental variable estimator is:

$$\hat{\mathbf{t}}_{\mathbf{bc-wiv}} = (\mathbf{G}_{\mathbf{bc}}'\mathbf{W}_{\mathbf{bc}}^{-1}\mathbf{A}_n)^{-1}\mathbf{G}_{\mathbf{bc}}'\mathbf{W}_{\mathbf{bc}}^{-1}\mathbf{B}_n. \tag{47}$$

Here

$$\frac{1}{n}\mathbf{G}_{\mathbf{bc}}^*\mathbf{W}_{\mathbf{bc}}^{-1}\mathbf{A}_n \approx \frac{\mathbb{E}(\cos e_i)}{n}\sum_{i=1}^{n}\begin{bmatrix} \dfrac{\sin^2\phi_i}{||\hat{\mathbf{t}}_{\mathbf{bc}} - \mathbf{s_i}||^2} & -\dfrac{\sin\phi_i\cos\phi_i}{||\hat{\mathbf{t}}_{\mathbf{bc}} - \mathbf{s_i}||^2} \\ -\dfrac{\sin\phi_i\cos\phi_i}{||\hat{\mathbf{t}}_{\mathbf{bc}} - \mathbf{s_i}||^2} & \dfrac{\cos^2\phi_i}{||\hat{\mathbf{t}}_{\mathbf{bc}} - \mathbf{s_i}||^2} \end{bmatrix}, \tag{48}$$

$$\frac{1}{n}\mathbf{G}_{\mathbf{bc}}^*\mathbf{W}_{\mathbf{bc}}^{-1}\mathbf{B}_n \approx \frac{\mathbb{E}(\cos e_i)}{n}\sum_{i=1}^{n}\begin{bmatrix} \dfrac{\sin^2\phi_i x_{s_i} - \sin\phi_i\cos\phi_i y_{s_i}}{||\hat{\mathbf{t}}_{\mathbf{bc}} - \mathbf{s_i}||^2} \\ \dfrac{-\sin\phi_i\cos\phi_i x_{s_i} + \cos^2\phi_i y_{s_i}}{||\hat{\mathbf{t}}_{\mathbf{bc}} - \mathbf{s_i}||^2} \end{bmatrix} \tag{49}$$

in probability as $n \to \infty$. Hence $\hat{\mathbf{t}}_{\mathbf{bc-wiv}}$ is an asymptotically unbiased estimator. Its covariance

$$
\begin{aligned}
&\mathbb{E}((\hat{\mathbf{t}}_{\mathbf{bc-wiv}} - \mathbf{t})(\hat{\mathbf{t}}_{\mathbf{bc-wiv}} - \mathbf{t})') \\
&\approx \left(\frac{1}{n}\mathbf{G}'_{\mathbf{bc}}\mathbf{W}_{\mathbf{bc}}^{-1}\mathbf{G}_{\mathbf{bc}}\right)^{-1}\left(\frac{1}{n^2}\mathbf{G}'_{\mathbf{bc}}\mathbf{W}_{\mathbf{bc}}^{-1}\mathbb{E}((\mathbf{B}_n - \mathbf{A}_n\mathbf{t})(\mathbf{B}_n - \mathbf{A}_n\mathbf{t})')\mathbf{W}_{\mathbf{bc}}^{-1}\mathbf{G}_{\mathbf{bc}}\right)\left(\frac{1}{n}\mathbf{G}'_{\mathbf{bc}}\mathbf{W}_{\mathbf{bc}}^{-1}\mathbf{G}_{\mathbf{bc}}\right)^{-1} \\
&\approx \frac{\mathbb{E}(\sin^2 e_i)}{n}\left(\frac{1}{n}\mathbf{G}'_{\mathbf{bc}}\mathbf{W}_{\mathbf{bc}}^{-1}\mathbf{G}_{\mathbf{bc}}\right)^{-1} \\
&\approx \mathbf{CRLB}
\end{aligned}
\tag{50}
$$

,

when $\mathbb{E}(\sin^2 e_i)$ is small as $n \to \infty$. In z-axis direction, by constructing matrix

$$
\mathbf{G}_{\mathbf{zbc}} = \begin{bmatrix} \cos\bar{\theta}_1 \\ \vdots \\ \cos\bar{\theta}_n \end{bmatrix}, \mathbf{W}_{\mathbf{zbc}} = \begin{bmatrix} ||\hat{\mathbf{T}}_{\mathbf{bc}} - \mathbf{S_1}||^2 & 0 & 0 \\ 0 & \ddots & 0 \\ 0 & 0 & ||\hat{\mathbf{T}}_{\mathbf{bc}} - \mathbf{S_n}||^2 \end{bmatrix},
\tag{51}
$$

where

$$
\bar{\theta}_i = \tan^{-1}\left(\frac{\hat{z}_{bc} - z_{s_i}}{\hat{r}_{bc-wiv}}\right),
\tag{52}
$$

$$
\hat{r}_{bc-wiv} = \sqrt{\left(\hat{\mathbf{t}}_{\mathbf{bc-wiv}}(1) - x_{s_i}\right)^2 + \left(\hat{\mathbf{t}}_{\mathbf{bc-wiv}}(2) - y_{s_i}\right)^2}.
\tag{53}
$$

The bias compensation weighted instrumental variable estimator in z axis is:

$$
\hat{z}_{bc-wiv} = (\mathbf{G}'_{\mathbf{zbc}}\mathbf{W}_{\mathbf{zbc}}^{-1}\mathbf{Y_n})^{-1}\mathbf{G}'_{\mathbf{zbc}}\mathbf{W}_{\mathbf{zbc}}^{-1}\mathbf{F}_n.
\tag{54}
$$

## 4. Sensor Trajectory Design

In this section we propose the one-step-ahead sensor maneuvering strategies for $\mathbf{S}_{k+1}(n \geq k \geq 3)$ based on $\mathbf{S}_1, \ldots, \mathbf{S}_k$ and discuss the properties of such strategies. It is worth pointing out beforehand that $\mathbf{S}_1, \ldots, \mathbf{S}_3$ are prefixed for motion model initialization and do not need to be designed.

### 4.1. Constraints

As shown in Figure 2, sensor trajectory planning is implemented under the following constraints [17,29]:

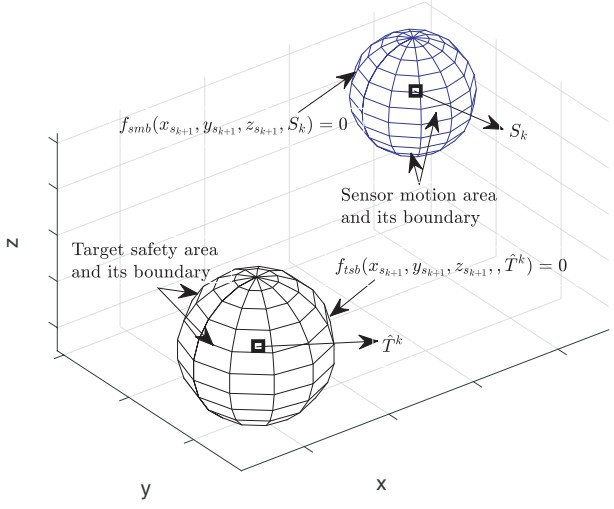

**Figure 2.** Sensor motion area and target safety area.

Target safety area: In order to ensure no collision between the sensor and the target, the distance between $\mathbf{S}_{k+1}$ and $\hat{\mathbf{T}}^k$ should be greater than or equal to the safe distance $R$.

Sensor motion area: Since the motion of the sensor is limited by the maximum speed, the sensor motion area is set to be spherical centering at $\mathbf{S}_k$ with the radius of $r$ (including the boundary of the sphere). Their boundaries are governed by polynomial equations $f_{tsb}(x_{\mathbf{s}_{k+1}}, y_{\mathbf{s}_{k+1}}, z_{\mathbf{s}_{k+1}}; \hat{\mathbf{T}}^k) = 0$ and $f_{smb}(x_{\mathbf{s}_{k+1}}, y_{\mathbf{s}_{k+1}}, z_{\mathbf{s}_{k+1}}; \mathbf{S}_k) = 0$, where

$$f_{tsb}(x_{\mathbf{s}_{k+1}}, y_{\mathbf{s}_{k+1}}, z_{\mathbf{s}_{k+1}}; \hat{\mathbf{T}}^k) = ||\mathbf{S}_{k+1} - \hat{\mathbf{T}}^k||^2 - R^2, \tag{55}$$

$$f_{smb}(x_{\mathbf{s}_{k+1}}, y_{\mathbf{s}_{k+1}}, z_{\mathbf{s}_{k+1}}; \mathbf{S}_k) = ||\mathbf{S}_{k+1} - \mathbf{S}_k||^2 - r^2. \tag{56}$$

### 4.2. Sensor Trajectory Planning Strategies

The position of sensor is deployed to improve the subsequent target positioning accuracy. According to [19], define:

$$\mathbf{M_1} = \frac{1}{\sigma^2} \sum_{i=1}^{k+1} \frac{1}{||\mathbf{t} - \mathbf{s}_i||^2} \begin{bmatrix} \sin^2\phi_i & -\frac{1}{2}\sin 2\phi_i & 0 \\ -\frac{1}{2}\sin 2\phi_i & \cos^2\phi_i & 0 \\ 0 & 0 & 0 \end{bmatrix}, \tag{57}$$

$$\mathbf{M_2} = \frac{1}{\zeta^2} \sum_{i=1}^{k+1} \frac{1}{||\mathbf{T} - \mathbf{S}_i||^2} \begin{bmatrix} \sin^2\theta_i\cos^2\phi_i & \frac{1}{2}\sin^2\theta_i\sin 2\phi_i & -\frac{1}{2}\sin 2\theta_i\cos\phi_i \\ \frac{1}{2}\sin^2\theta_i\sin 2\phi_i & \sin^2\theta_i\sin^2\phi_i & -\frac{1}{2}\sin 2\theta_i\sin\phi_i \\ -\frac{1}{2}\sin 2\theta_i\cos\phi_i & -\frac{1}{2}\sin 2\theta_i\sin\phi_i & \cos^2\theta_i \end{bmatrix}, \tag{58}$$

where the matrix $\mathbf{M_1}$ represents the azimuth FIM over $k+1$ measurements, as it corresponds to the 2D projection of the localization problem onto $xy$ plane, and the matrix $\mathbf{M_2}$ represents the elevation FIM over $k+1$ measurements. Therefore, the FIM over $k+1$ measurements in 3D space is:

$$\mathbf{FIM_{k+1}} = \mathbf{M_1} + \mathbf{M_2}. \tag{59}$$

**Lemma 3.**

$$\det(\mathbf{FIM_{k+1}}) = \sum_{\{p_1,\, p_2,\, p_3\}}^{\binom{2(k+1)}{3}} \det(\mathbf{V_{p_1}} + \mathbf{V_{p_2}} + \mathbf{V_{p_3}}), \tag{60}$$

*where the set $\{p_1,\ p_2,\ p_3\}$ is the combination selected from $\{1,\ \dots, 2(k+1)\}$.*

$\mathbf{V_{2i-1}}$ and $\mathbf{V_{2i}}$ $(i = 1, 2, 3, ..., k+1)$ are the azimuth FIM and the elevation FIM, respectively.

**Proof of Lemma 3.** By the definition of matrix determinant, we have:

$$\det(\mathbf{FIM_{k+1}}) = \sum_{\{p_1,\, p_2,\, p_3\}}^{\binom{2(k+1)}{3}} \det(\mathbf{V_{p_1}} + \mathbf{V_{p_2}} + \mathbf{V_{p_3}})$$

$$+ \Psi_1 \sum_{\{p_4,\, p_5\}}^{\binom{2(k+1)}{2}} \det(\mathbf{V_{p_4}} + \mathbf{V_{p_5}}) + \Psi_2 \sum_{\{p_6\}}^{\binom{2(k+1)}{1}} \det(\mathbf{V_{p_6}}), \tag{61}$$

where the set $\{p_4,\ p_5\}$ and $\{p_6\}$ are the combinations selected from $\{1,\ \dots, 2(k+1)\}$. $\Psi_1$ and $\Psi_2$ are constant factors decided by the combination numbers. Under $\mathrm{rank}(\mathbf{V_{p_4}} + \mathbf{V_{p_5}}) \leq 2$ and $\mathrm{rank}(\mathbf{V_{p_6}}) = 1$, $\det(\mathbf{V_{p_4}} + \mathbf{V_{p_5}}) = \det(\mathbf{V_{p_6}}) = 0$. Then (61) turns into (60). $\quad\square$

**Theorem 3.**

$$\det(\mathbf{FIM_{k+1}}) = \det(\mathbf{FIM_k}) + \sum_{\{p_7,\, p_8\}}^{\binom{2k}{2}} \det(\mathbf{V_{2k+1}} + \mathbf{V_{p_7}} + \mathbf{V_{p_8}}) \tag{62}$$

$$+ \sum_{\{p_9,\, p_{10}\}}^{\binom{2k+1}{2}} \det(\mathbf{V_{2(k+1)}} + \mathbf{V_{p_9}} + \mathbf{V_{p_{10}}}), \tag{63}$$

*where the set* $\{p_7,\ p_8\}$ *and* $\{p_9,\ p_{10}\}$ *are the combination selected from* $\{1,\dots,2k\}$ *and* $\{1,\dots,2(k+1)\}$.

**Proof of Theorem 3.** From Lemma 3,

$$\det(\mathbf{FIM_k}) = \sum_{\{p_{11},\, p_{12},\, p_{13}\}}^{\binom{2k}{3}} \det(\mathbf{V_{p_{11}}} + \mathbf{V_{p_{12}}} + \mathbf{V_{p_{13}}}), \tag{64}$$

where the set $\{p_{11},\ p_{12},\ p_{13}\}$ is the combination selected from $\{1,\ \dots\ 2k\}$.

$$\det(\mathbf{FIM_{k+1}}) - \det(\mathbf{FIM_k}) = \sum_{\{p_7,\, p_8\}}^{\binom{2k}{2}} \det(\mathbf{V_{2k+1}} + \mathbf{V_{p_7}} + \mathbf{V_{p_8}}) \tag{65}$$

$$+ \sum_{\{p_9,\, p_{10}\}}^{\binom{2k+1}{2}} \det(\mathbf{V_{2(k+1)}} + \mathbf{V_{p_9}} + \mathbf{V_{p_{10}}}) \tag{66}$$

naturally follows. □

Let the location $\mathbf{S}_{k+1}$ on the trajectory is generated pointwisely through the maximization of

$$\max_{\mathbf{S_4},\dots \mathbf{S_n}} \det(\mathbf{FIM_n}) \geq \sum_{k=4}^{n} \max_{\mathbf{S_{k+1}}}(\det(\mathbf{FIM_{k+1}}) - \det(\mathbf{FIM_k})) + \det(\mathbf{FIM_k}), \tag{67}$$

which shows the trajectory is suboptimal. According to Theorem 3, it is still a reasonable choice because the design of $\mathbf{S}_{k+1}$ considers information given by $\mathbf{S}_{k+1}$ and all the past sensor locations jointly. Defining

$$f(\mathbf{S}_{k+1}) = \det(\mathbf{FIM_{k+1}}) - \det(\mathbf{FIM_k}) \tag{68}$$

as the objective function of solving $\mathbf{S_{k+1}}$. Formally defining

**Trajectory 1.** *the location* $\mathbf{S}_{k+1}$ *on the trajectory is generated pointwisely through the maximization of*

$$\max_{\mathbf{S_{k+1}}} f(\mathbf{S}_{k+1}) \tag{69}$$

$$s.t.\ ||\mathbf{S}_{k+1} - \mathbf{S}_k|| \leqslant r, \tag{70}$$

$$||\mathbf{S}_{k+1} - \mathbf{T}^k|| \geqslant R \tag{71}$$

*with respect to* $\mathbf{S}_{k+1}$ *given* $\mathbf{S}_1,\dots,\mathbf{S}_k(k \geqslant 3)$.

When optimizing $f(\mathbf{S}_{k+1})$, unknown $\mathbf{T}$, $\phi_i$, $\theta_i$, $\sigma^2$ and $\xi^2$ $(i = 1,\dots,k)$ are replaced by the estimated $\hat{\mathbf{T}}^k$, $\hat{\phi}_i(\hat{\mathbf{T}}^k)$ and $\hat{\theta}_i(\hat{\mathbf{T}}^k)$, $\hat{\sigma}^2(\hat{\mathbf{T}}^k)$ and $\hat{\xi}^2(\hat{\mathbf{T}}^k)$. This method updates target location and re-estimates all the historical data in each step. To circumvent this weakness, we present

**Trajectory 2.** *the location $\mathbf{S}_{k+1}$ on the trajectory is generated pointwisely through the maximization of*

$$\max_{\mathbf{S}_{k+1}} \quad \hat{f}(\mathbf{S}_{k+1}) \tag{72}$$

$$s.t. \; ||\mathbf{S}_{k+1} - \mathbf{S}_k|| \leqslant r, \tag{73}$$

$$||\mathbf{S}_{k+1} - \mathbf{T}^k|| \geqslant R \tag{74}$$

*with respect to $\mathbf{S}_{k+1}$ given $\mathbf{S}_1, \ldots, \mathbf{S}_k (k \geqslant 3)$.*

When optimizing $\hat{f}(\mathbf{S}_{k+1})$, $\sigma^2$ and $\xi^2$ are replaced by $(\sum_{i=1}^{k} \hat{\sigma}_i^2(\hat{\mathbf{T}}^{i-1}) + \hat{\sigma}_{k+1}^2(\hat{\mathbf{T}}^k))/(k+1)$ and $(\sum_{i=1}^{k} \hat{\xi}_i^2(\hat{\mathbf{T}}^{i-1}) + \hat{\xi}_{k+1}^2(\hat{\mathbf{T}}^k))/(k+1)$, respectively. Distinct with trajectory 1, $\hat{\phi}_i$ and $\hat{\theta}_i$ $(i = 1, \ldots, k)$ are not updated by $\hat{\mathbf{T}}^k$.

In Trajectory 1 and 2, the computation time to generate the sensor position at the $(k+1)$th moment includes the time to obtain the objective function and the time to optimize the objective function. Trajectory 1 needs $(k+1)*(49$ additions/multiplications, three square roots, nine trigonometric function calculations) and 18 additions/multiplications for the calculation of $\det(\mathbf{FIM_{k+1}})$ to obtain the objective function, while trajectory 2 requires only 49 additions/multiplications, three square roots, nine trigonometric functions and 18 additions/multiplications for the calculation of $\det(\mathbf{FIM_{k+1}})$. The generation of trajectory 2 reduces $k * (49$ additions/multiplications, three square root, nine trigonometric function calculations) calculations compared to trajectory 1. In addition, the objective function structure is similar, the optimization time of the objective function is almost equal depending on the optimization method.

*4.3. Optimal Solution Region*

Through the analysis of (62) , we have:

**Theorem 4.**

$$\det(\mathbf{FIM_{k+1}}) - \det(\mathbf{FIM_k}) = \sum_{\substack{c_i \in \{p_7, \, p_8, \, p_9\} \\ c_i \notin \{p_{13}, \, p_{14}\}}} \sum_{j_i \in \{1, \ldots 3\}} \left( \prod_{i=1}^{3} sgn(j) L_{i,j_i}^{(c_i)} \right), \tag{75}$$

*where the set $\{p_7, \, p_8, \, p_9\}$ is the combination selected from $\{1, \ldots 2(k+1)\}$ with $\exists p_j (7 \leq j \leq 9) \in \{2k+1, 2(k+1)\}$. The set $\{p_{13}, \, p_{14}\}$ is the combination selected from $\{p_7, \, p_8, \, p_9\}$. $j$ represents the permutation of $\{j_1, j_2, j_3\}$. The sign function $sgn(j) = -1$ when the number of exchanges that $\{j_1, j_2, j_3\}$ takes to be the standard order $\{1, 2, 3\}$ is odd. Otherwise, $sgn(j) = 1$. $L_{i,j_i}^{(c_i)}$ is the ith row and $j_i$th column element in $V_{c_i}$.*

**Proof of Theorem 4.** From Theorem 3 and the Leibniz formula,

$$\det(\mathbf{FIM_{k+1}}) - \det(\mathbf{FIM_k}) = \sum_{c_i \in \{p_7, \, p_8, \, p_9\}} \sum_{j_i \in \{1, \ldots 3\}} \left( \prod_{i=1}^{3} \text{sgn}(j) L_{i,j_i}^{(c_i)} \right). \tag{76}$$

In addition,

$$\sum_{\{p_{13}, \, p_{14}\}} \det(\mathbf{V_{p_{13}}} + \mathbf{V_{p_{14}}}) = \sum_{c_i \in \{p_{13}, \, p_{14}\}} \sum_{j_i \in \{1, \ldots 3\}} \left( \prod_{i=1}^{3} \text{sgn}(j) L_{i,j_i}^{(c_i)} \right) = 0. \tag{77}$$

Deducting (77) from (76) leads to (75).  □

From Theorem 4, (62) can be rewritten uniformly into:

$$f(\mathbf{S}_{k+1}) = \frac{\displaystyle\sum_{h=0}^{4}\sum_{q=0}^{4}\sum_{l=0}^{2} \alpha_{hql} x_{\mathbf{s}_{k+1}}^{h} y_{\mathbf{s}_{k+1}}^{q} z_{\mathbf{s}_{k+1}}^{l}}{\displaystyle\sum_{h^+=0}^{10}\sum_{q^+=0}^{10}\sum_{l^+=0}^{6} \beta_{h^++q^++l^+} x_{\mathbf{s}_{k+1}}^{h^+} y_{\mathbf{s}_{k+1}}^{q^+} z_{\mathbf{s}_{k+1}}^{l^+}}, \tag{78}$$

where the coefficient $\alpha_{pql}$ and $\beta_{p^++q^++l^+}$ can be zero. According to Theorem 4, when the terms $2k+1$ and $2(k+1)$ in (75) are selected in set $\{p_7,\ p_8,\ p_9\}$, the numerator and denominator of $f(\mathbf{S}_{k+1})$ have the highest order with respect to $x$, $y$ and $z$. Since $\mathbf{V}_{2k+1}$ and $\mathbf{V}_{2(k+1)}$ contain $x$, $y$ and $z$, when only the term $2(k+1)$ in (75) is selected in set $\{p_7,\ p_8,\ p_9\}$, neither of the numerator and denominator in (78) has the highest order. To make $\mathbf{V}_{2k+1}$ and $\mathbf{V}_{2(k+1)}$ in matrix form with $x_{\mathbf{s}_{k+1}}, y_{\mathbf{s}_{k+1}}, z_{\mathbf{s}_{k+1}}$, we get the expression:

$$\mathbf{V}_{2k+1} = \frac{1}{\sigma_{k+1}^2} \frac{1}{||\hat{\mathbf{t}}^k - \mathbf{s}_{k+1}||^2}$$

$$\begin{bmatrix} \dfrac{(y_{\mathbf{s}_{k+1}}-\hat{\mathbf{T}}^k(2))^2}{||\hat{\mathbf{t}}^k-\mathbf{s}_{k+1}||^2} & -\dfrac{(x_{\mathbf{s}_{k+1}}-\hat{\mathbf{T}}^k(1))(y_{\mathbf{s}_{k+1}}-\hat{\mathbf{T}}^k(2))}{||\hat{\mathbf{t}}^k-\mathbf{s}_{k+1}||^2} & 0 \\[2em] -\dfrac{(x_{\mathbf{s}_{k+1}}-\hat{\mathbf{T}}^k(1))(y_{\mathbf{s}_{k+1}}-\hat{\mathbf{T}}^k(2))}{||\hat{\mathbf{t}}^k-\mathbf{s}_{k+1}||^2} & \dfrac{(x_{\mathbf{s}_{k+1}}-\hat{\mathbf{T}}^k(1))^2}{||\hat{\mathbf{t}}^k-\mathbf{s}_{k+1}||^2} & 0 \\[2em] 0 & 0 & 0 \end{bmatrix}, \tag{79}$$

$$\mathbf{V}_{2(k+1)} = \frac{1}{\varsigma_{k+1}^2} \frac{1}{||\hat{\mathbf{T}}^k - \mathbf{S}_{k+1}||^2}$$

$$\begin{bmatrix} \dfrac{(z_{\mathbf{s}_{k+1}}-\hat{\mathbf{T}}^k(3))^2(x_{\mathbf{s}_{k+1}}-\hat{\mathbf{T}}^k(1))^2}{||\hat{\mathbf{T}}^k-\mathbf{S}_{k+1}||^2||\hat{\mathbf{t}}^k-\mathbf{s}_{k+1}||^2} & \dfrac{(z_{\mathbf{s}_{k+1}}-\hat{\mathbf{T}}^k(3))^2(x_{\mathbf{s}_{k+1}}-\hat{\mathbf{T}}^k(1))(y_{\mathbf{s}_{k+1}}-\hat{\mathbf{T}}^k(2))}{||\hat{\mathbf{T}}^k-\mathbf{S}_{k+1}||^2||\hat{\mathbf{t}}^k-\mathbf{s}_{k+1}||^2} & -\dfrac{(z_{\mathbf{s}_{k+1}}-\hat{\mathbf{T}}^k(3))(x_{\mathbf{s}_{k+1}}-\hat{\mathbf{T}}^k(1))}{||\hat{\mathbf{T}}^k-\mathbf{S}_{k+1}||^2} \\[2em] \dfrac{(z_{\mathbf{s}_{k+1}}-\hat{\mathbf{T}}^k(3))^2(x_{\mathbf{s}_{k+1}}-\hat{\mathbf{T}}^k(1))(y_{\mathbf{s}_{k+1}}-\hat{\mathbf{T}}^k(2))}{||\hat{\mathbf{T}}^k-\mathbf{S}_{k+1}||^2||\hat{\mathbf{t}}^k-\mathbf{s}_{k+1}||^2} & \dfrac{(z_{\mathbf{s}_{k+1}}-\hat{\mathbf{T}}^k(3))^2(y_{\mathbf{s}_{k+1}}-\hat{\mathbf{T}}^k(2))^2}{||\hat{\mathbf{T}}^k-\mathbf{S}_{k+1}||^2||\hat{\mathbf{t}}^k-\mathbf{s}_{k+1}||^2} & -\dfrac{(z_{\mathbf{s}_{k+1}}-\hat{\mathbf{T}}^k(3))(y_{\mathbf{s}_{k+1}}-\hat{\mathbf{T}}^k(2))}{||\hat{\mathbf{T}}^k-\mathbf{S}_{k+1}||^2} \\[2em] -\dfrac{(z_{\mathbf{s}_{k+1}}-\hat{\mathbf{T}}^k(3))(x_{\mathbf{s}_{k+1}}-\hat{\mathbf{T}}^k(1))}{||\hat{\mathbf{T}}^k-\mathbf{S}_{k+1}||^2} & -\dfrac{(z_{\mathbf{s}_{k+1}}-\hat{\mathbf{T}}^k(3))(y_{\mathbf{s}_{k+1}}-\hat{\mathbf{T}}^k(2))}{||\hat{\mathbf{T}}^k-\mathbf{S}_{k+1}||^2} & \dfrac{||\hat{\mathbf{t}}^k-\mathbf{s}_{k+1}||^2}{||\hat{\mathbf{T}}^k-\mathbf{S}_{k+1}||^2}. \end{bmatrix}. \tag{80}$$

Take the highest order of $x_{\mathbf{s}_{k+1}}$ as an example. Only when the item $(x_{\mathbf{s}_{k+1}} - \hat{\mathbf{T}}^k(1))^2/||\hat{\mathbf{t}}^k-\mathbf{s}_{k+1}||^2$ in $\mathbf{V}_{2k+1}$ and the item $(z_{\mathbf{s}_{k+1}} - \hat{\mathbf{T}}^k(3))^2(x_{\mathbf{s}_{k+1}} - \hat{\mathbf{T}}^k(1))^2/||\hat{\mathbf{T}}^k - \mathbf{S}_{k+1}||^2||\hat{\mathbf{t}}^k-\mathbf{s}_{k+1}||^2$ in $\mathbf{V}_{2(k+1)}$ are selected, the numerator and denominator of $f(\mathbf{S}_{k+1})$ have the highest order with respect to $x$, which are the 4th order and the 10th order, respectively.

$\bar{\Omega} = \Omega \cup \partial\Omega$ is a compact and connected set in 3D space which $\Omega$ stands for all internal points of the feasible area, $\partial\Omega$ represents the boundary of the feasible area. When the target point $\hat{\mathbf{T}}^k$ is outside the sensor motion area, make tangent lines from point $\hat{\mathbf{T}}^k$ to the sensor motion area and all tangent points can form a circle. Denote the cone formed by vertex $\hat{\mathbf{T}}^k$ and the circle as $\triangle_{TAB}$ in Figure 3 (the tangent points $A$ and $B$ are symmetrically distributed).

In order to solve the proposed optimization problem quickly, according to the relative position of the sensor motion area and the target safety area, the feasible region of the sensor trajectory optimization problem can be analyzed in following cases.

1.  When the sensor motion area and the target safety area are separated, define the boundary part of the sensor motion area inside $\triangle_{TAB}$ as surface 1 in Figure 3a. The feasible region is the sensor motion area;

2.  When the sensor motion area intersects with the target safety area, the feasible region is the sensor motion area where the part inside the target safety area is excluded, and the surface is where part of the target safety boundary is inside the sensor motion area. According to the position of $\hat{\mathbf{T}}^k$, it can be divided into three sub-cases: (i) when

$\hat{\mathbf{T}}^k$ is outside the sensor motion area and $\triangle_{TAB}$ is not contained by the target safety area, define the part of the sensor motion boundary inside $\triangle_{TAB}$ with the section inside the target safety area replaced by the section of the target safety boundary inside the sensor movement area, such as surface 2 in Figure 3b. *E* and *F* (*E* and *F* are symmetrically distributed) are the points where the two spheres intersect; (ii) when the $\hat{\mathbf{T}}^k$ is outside the sensor motion area and if $\triangle_{TAB}$ is contained by the target safety area, define the part of the target safety boundary inside the sensor motion area as surface 3 in Figure 3c; (iii) when the $\hat{\mathbf{T}}^k$ is inside the sensor motion area, define the part of the target safety boundary inside the sensor motion area as surface 4 in Figure 3d;

3. when the sensor motion area contains the target safety area, the feasible region is the sensor motion area where the interior section of the target safety area is excluded. The target safety boundary is defined as surface 5 in Figure 3e.

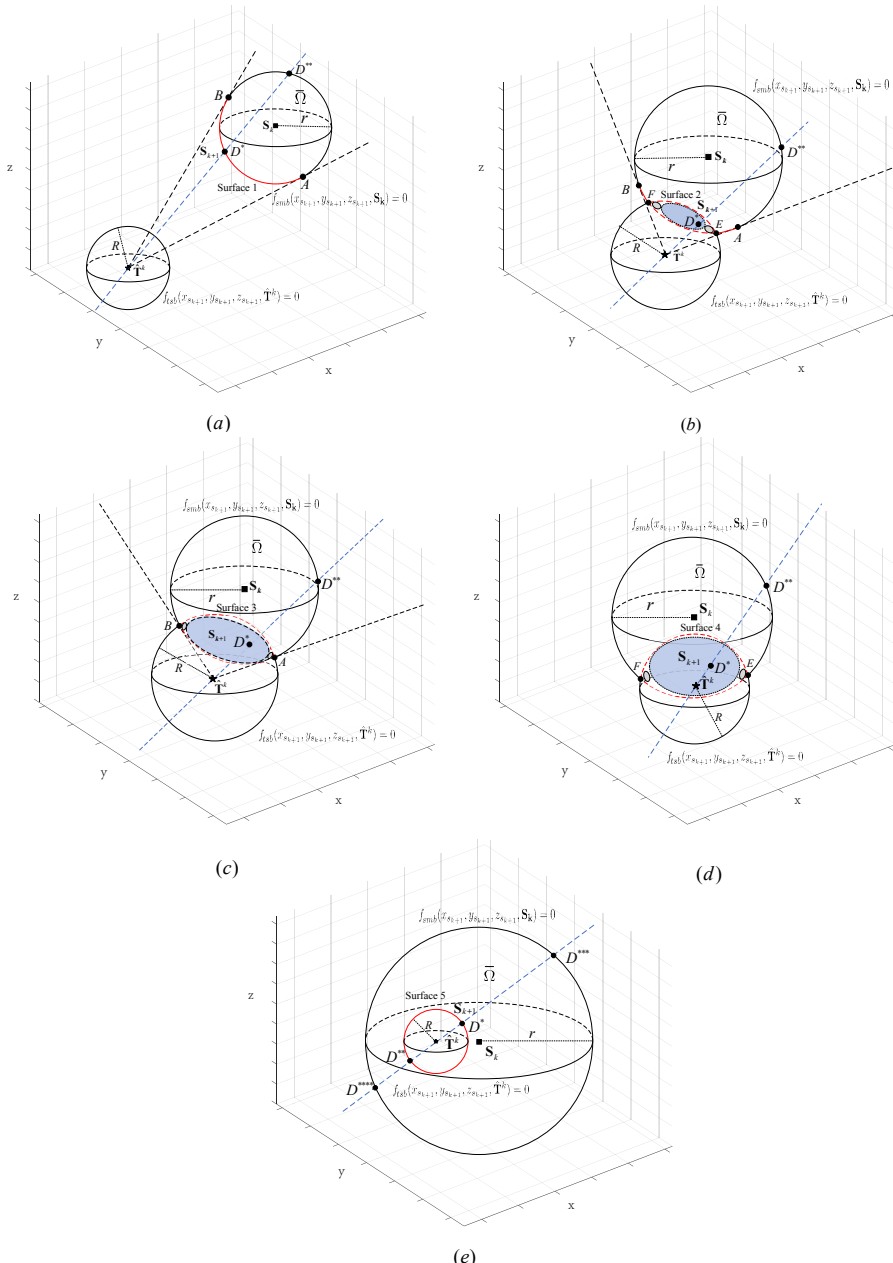

**Figure 3.** Surface 1 (red part in (**a**)) when the two areas are separated, surface 2 (red part in (**b**)), surface 3 (red part in (**c**)) and surface 4 (red part in (**d**)) when the two areas intersect, the target safety boundary (red part in (**e**)) when the target safety area is in the sensor movement area.

**Proposition 1.** *In case 1, the maximizer of the $f(\mathbf{S}_{k+1})$ exists on the spherical crown (surface 1). In case 2(i), the maximizer of the $f(\mathbf{S}_{k+1})$ exists on surface 2 where the two areas intersect in the sensor motion area, and the side of the circular truncated cone (lower bottom circle with AB as the diameter and upper bottom circle with EF as the diameter). In cases 2(ii) and 2(iii), the maximizer of the $f(\mathbf{S}_{k+1})$ exists on surfaces 3, 4 where the two areas intersect. In case 3, the maximizer of $f(\mathbf{S}_{k+1})$ is on the boundary of the target safety area (surface 5).*

**Proof of Proposition 1.** Regardless of the relative location of the two areas, a straight blue dotted line is drawn between $\hat{\mathbf{T}}^k$ and any point in the feasible region; all the points on this line have the same $\hat{\phi}_{k+1}(\hat{\mathbf{T}}^k)$ and $\hat{\theta}_{k+1}(\hat{\mathbf{T}}^k)$ as shown. Therefore the maximizer of $f(\mathbf{S}_{k+1})$ on the line segment is the closest one $D^*$ to $\hat{\mathbf{T}}^k$ in the Euclidean sense. Gathering up these maximizers gives different surfaces such as the red part in Figure 3.  □

*4.4. Analytical Derivation for the Global Maximizer*

According to (78), the partial derivative of $f(\mathbf{S}_{k+1})$ over $x_{\mathbf{s}_{k+1}}$, $y_{\mathbf{s}_{k+1}}$ and $z_{\mathbf{s}_{k+1}}$ can be written into:

$$\frac{\partial f(\mathbf{S}_{k+1})}{\partial x_{\mathbf{s}_{k+1}}} = \frac{\sum_{h=0}^{13}\sum_{q=0}^{14}\sum_{l=0}^{8}\alpha_{hql}^1(y_{\mathbf{s}_{k+1}}^q, z_{\mathbf{s}_{k+1}}^l)x_{\mathbf{s}_{k+1}}^h}{\sum_{h^+=0}^{20}\sum_{q^+=0}^{20}\sum_{l^+=0}^{12}\beta_{h^+q^+l^+}^+ x_{\mathbf{s}_{k+1}}^{h^+} y_{\mathbf{s}_{k+1}}^{q^+} z_{\mathbf{s}_{k+1}}^{l^+}}, \tag{81}$$

$$\frac{\partial f(\mathbf{S}_{k+1})}{\partial y_{\mathbf{s}_{k+1}}} = \frac{\sum_{h=0}^{14}\sum_{q=0}^{13}\sum_{l=0}^{8}\alpha_{hql}^2(y_{\mathbf{s}_{k+1}}^q, z_{\mathbf{s}_{k+1}}^l)x_{\mathbf{s}_{k+1}}^h}{\sum_{h^+=0}^{20}\sum_{q^+=0}^{20}\sum_{l^+=0}^{12}\beta_{h^+q^+l^+}^+ x_{\mathbf{s}_{k+1}}^{h^+} y_{\mathbf{s}_{k+1}}^{q^+} z_{\mathbf{s}_{k+1}}^{l^+}}, \tag{82}$$

$$\frac{\partial f(\mathbf{S}_{k+1})}{\partial z_{\mathbf{s}_{k+1}}} = \frac{\sum_{h=0}^{14}\sum_{q=0}^{14}\sum_{l=0}^{7}\alpha_{hql}^3(y_{\mathbf{s}_{k+1}}^q, z_{\mathbf{s}_{k+1}}^l)x_{\mathbf{s}_{k+1}}^h}{\sum_{h^+=0}^{20}\sum_{q^+=0}^{20}\sum_{l^+=0}^{12}\beta_{h^+q^+l^+}^+ x_{\mathbf{s}_{k+1}}^{h^+} y_{\mathbf{s}_{k+1}}^{q^+} z_{\mathbf{s}_{k+1}}^{l^+}}, \tag{83}$$

where the coefficient $\alpha_{hql}^1$, $\alpha_{hql}^2$, $\alpha_{hql}^3$ and $\beta_{h^+q^+l^+}^+$ can be zero. From Proposition 1, the feasible region always lies on the area boundary. Take the case when the two areas are separated as an example. $[\partial f(\mathbf{S}_{k+1})/\partial x_{s_{k+1}} \quad \partial f(\mathbf{S}_{k+1})/\partial y_{s_{k+1}} \quad \partial f(\mathbf{S}_{k+1})/\partial z_{s_{k+1}}]'$ should be colinear with the normal vector $[x_{\mathbf{s}_{k+1}} - x_{\mathbf{s}_k} \quad y_{\mathbf{s}_{k+1}} - y_{\mathbf{s}_k} \quad z_{\mathbf{s}_{k+1}} - z_{\mathbf{s}_k}]'$ at the optimal, i.e.,

$$\frac{\partial f(\mathbf{S}_{k+1})}{\partial x_{\mathbf{s}_{k+1}}}(y_{\mathbf{s}_{k+1}} - y_{\mathbf{s}_k}) - \frac{\partial f(\mathbf{S}_{k+1})}{\partial y_{\mathbf{s}_{k+1}}}(x_{\mathbf{s}_{k+1}} - x_{\mathbf{s}_k}) = 0, \tag{84}$$

$$\frac{\partial f(\mathbf{S}_{k+1})}{\partial x_{\mathbf{s}_{k+1}}}(z_{\mathbf{s}_{k+1}} - z_{\mathbf{s}_k}) - \frac{\partial f(\mathbf{S}_{k+1})}{\partial z_{\mathbf{s}_{k+1}}}(x_{\mathbf{s}_{k+1}} - x_{\mathbf{s}_k}) = 0. \tag{85}$$

Otherwise, any infinitesimal move along the tangent vector that constructs an obtuse angle with $-[\partial f(\mathbf{S}_{k+1})/\partial x_{\mathbf{s}_{k+1}} \quad \partial f(\mathbf{S}_{k+1})/\partial y_{\mathbf{s}_{k+1}} \quad \partial f(\mathbf{S}_{k+1})/\partial z_{\mathbf{s}_{k+1}}]'$ increases the value of $f(\mathbf{S}_{k+1})$, (84) is equivalent to:

$$\sum_{h=0}^{15} g_h(y_{\mathbf{s}_{k+1}}, z_{\mathbf{s}_{k+1}})x_{\mathbf{s}_{k+1}}^h = 0, \tag{86}$$

where $g_h(y_{\mathbf{s}_{k+1}}, z_{\mathbf{s}_{k+1}})$ is the polynomial of $y_{\mathbf{s}_{k+1}}, z_{\mathbf{s}_{k+1}}$. Define $\eta(y_{\mathbf{s}_{k+1}}, z_{\mathbf{s}_{k+1}}) = (y_{\mathbf{s}_{k+1}} - y_{\mathbf{s}_k})^2 + (z_{\mathbf{s}_{k+1}} - z_{\mathbf{s}_k})^2 + x_{\mathbf{s}_k}^2 - r^2$. The movement boundary condition is rewritten into:

$$\eta(y_{\mathbf{s}_{k+1}}, z_{\mathbf{s}_{k+1}}) - 2x_{\mathbf{s}_k}x_{\mathbf{s}_{k+1}} + x_{\mathbf{s}_{k+1}}^2 = 0. \tag{87}$$

The joint solution of (84) and (87) can be developed as follows. The $(15+2) \times (15+2)$ Sylvester matrix of (84) and (87) with respect to $y_{\mathbf{s}_{k+1}}, z_{\mathbf{s}_{k+1}}$ is:

$$\mathbf{Syl_1}((84),(87);y_{\mathbf{s}_{k+1}},z_{\mathbf{s}_{k+1}}) =$$
$$\begin{pmatrix} g_{15}(y_{\mathbf{s}_{k+1}},z_{\mathbf{s}_{k+1}}) & 0 & 1 & 0 & \dots & 0 \\ g_{14}(y_{\mathbf{s}_{k+1}},z_{\mathbf{s}_{k+1}}) & g_{15}(y_{\mathbf{s}_{k+1}},z_{\mathbf{s}_{k+1}}) & -2x_{\mathbf{s}_k} & 1 & \dots & 0 \\ g_{13}(y_{\mathbf{s}_{k+1}},z_{\mathbf{s}_{k+1}}) & g_{14}(y_{\mathbf{s}_{k+1}},z_{\mathbf{s}_{k+1}}) & \eta(y_{\mathbf{s}_{k+1}},z_{\mathbf{s}_{k+1}}) & -2x_{\mathbf{s}_k} & \dots & 0 \\ \vdots & \vdots & \vdots & \vdots & \vdots & \vdots \\ g_1(y_{\mathbf{s}_{k+1}},z_{\mathbf{s}_{k+1}}) & g_2(y_{\mathbf{s}_{k+1}},z_{\mathbf{s}_{k+1}}) & 0 & 0 & \dots & 1 \\ g_0(y_{\mathbf{s}_{k+1}},z_{\mathbf{s}_{k+1}}) & g_1(y_{\mathbf{s}_{k+1}},z_{\mathbf{s}_{k+1}}) & 0 & 0 & \dots & -2x_{\mathbf{s}_k} \\ 0 & g_0(y_{\mathbf{s}_{k+1}},z_{\mathbf{s}_{k+1}}) & 0 & 0 & \dots & \eta(y_{\mathbf{s}_{k+1}},z_{\mathbf{s}_{k+1}}) \end{pmatrix}. \quad (88)$$

The Sylvester resultant $\det(\mathbf{Syl_1}((84),(87);y_{\mathbf{s}_{k+1}},z_{\mathbf{s}_{k+1}}))$ is a bivariate polynomial with an upper bound of the 58th order for $y_{\mathbf{s}_{k+1}}$ and the 46th order for $z_{\mathbf{s}_{k+1}}$.

Similarly, the joint solution of (85) and (87) can be developed as the $(15+2) \times (15+2)$ Sylvester matrix of (85) and (87) is similar to (88). $\det(\mathbf{Syl_2}((85),(87);y_{\mathbf{s}_{k+1}},z_{\mathbf{s}_{k+1}}))$ is also a bivariate polynomial with an upper bound of the 58th order for $y_{\mathbf{s}_{k+1}}$ and the 46th order for $z_{\mathbf{s}_{k+1}}$. The real solution can be obtained by solving the equations given by the determinants of the two Sylvester matrices.

Comparing only the real solutions lying on surface 1 in terms of $f(\mathbf{S}_{k+1})$ gives the global maximizer. For surface 2, the real solutions of the surface equation and (86) are compared with the two intersection points to generate the global maximizer. For surfaces 3 and 4, the real solutions of the surface equation and (86) on the surface are compared with the two intersection points to provide $\mathbf{S}_{k+1}$. When the movement area contains the safety area, the real solutions of the safety constraint and the (86) that has the biggest $f(\mathbf{S}_{k+1})$ are the global maximizer.

**Remark 1.** *The position of a sensor should not be collinear with the target position in the z-axis direction so that the target is observable. It is necessary to set the cylindrical as a non-flying area in Figure 4. It means to insert a cylinder centering at the target position with the radius of $\rho$ in the target safety area, and the z-axis direction extends to infinity. The constraint is formulated as:*

$$(x_{k+1} - \hat{\mathbf{T}}^k(1))^2 + (y_{k+1} - \hat{\mathbf{T}}^k(2))^2 \geqslant \rho^2. \quad (89)$$

*If the maximum solution of $f(\mathbf{S}_{k+1})$ falls into the section where the surface intersects the cylinder, we choose the other one which leads to the second largest $f(\mathbf{S}_{k+1})$, and so forth. If all the maximum solutions of $f(\mathbf{S}_{k+1})$ fall within the intersection section, a point is randomly selected from the feasible region as the solution of $f(\mathbf{S}_{k+1})$.*

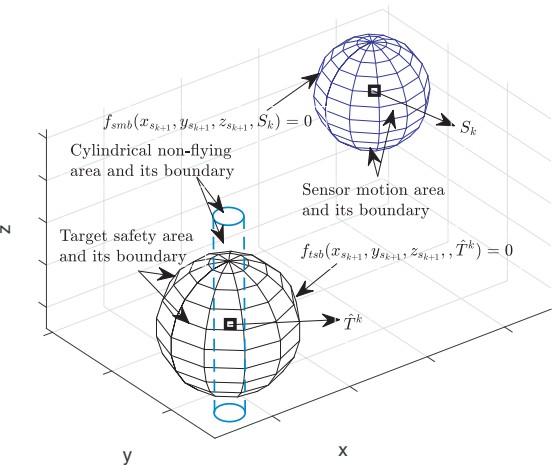

**Figure 4.** Sensor motion area, target safety area and cylindrical non-flying area.

*4.5. Single Sensor Trajectory Design Procedure*

We summarize the sensor trajectory design procedure in the following algorithm.

Trajectory design based localization algorithm:

Initialization: Use the initial position of the sensor to generate the initial estimator parameter by localization algorithm, e.g., Kalman-related methods.

Repeat until n is reached:

1.  Generate $\mathbf{S}_{k+1}$ through the maximization of (69) or (72). The maximizer of (69) or (72) can be produced in an analytical way described in Section 4.4;
2.  Gather the measurement information $\hat{\phi}_{k+1}$, $\hat{\theta}_{k+1}$, $\hat{\sigma}^2$ and $\hat{\xi}^2$ from the sensor at $\mathbf{S}_{k+1}$;
3.  Use localization technique to generate $\hat{\mathbf{T}}^{k+1}$;
4.  $k = k + 1$.

Output the target location.

In many applications, various location-based services need to determine accurate positioning information, such as for public safety services, field rescue and other fields. It is necessary to track the target under motion constraints. Unlike static sensors, which have a fixed density and perception range, mobile sensors can cover a larger area over time without increasing their number. In addition, their spatial distribution can change dynamically to accommodate the movement of the target, thus providing measurements of information about its position. Choosing the best sensing location is particularly important, especially given time-constrained applications such as tracking hostile targets. For example, autonomous underwater vehicles (AUV) are underwater unmanned autonomous vehicle platforms without a cable connection that can be applied for underwater environment monitoring, offshore oil engineering operations, underwater search and mapping after loading appropriate sensors. In the main-slave UUVS (Unmanned Underwater Vehicles) system, the slave-UUV only carries the azimuth information of the main-UUV under a cluttered environment and realizes the convergence with the main-UUV as soon as possible through autonomous optimization of its movement trajectory. The localization method and trajectory planning method proposed in this paper can be applied to the above scenarios.

## 5. Simulation

In this section, the bias compensation method and sensor trajectory planning algorithm are compared with other benchmarks through MATLAB R2020a simulation. The location unit used for the following experiments is a metre.

*5.1. Example 1*

We now explore the positioning performance of the proposed method and estimate the deviation through simulation. The simulation is performed in 3D space, where the target is assumed to be at position $\mathbf{T} = \begin{bmatrix} 30 & 40 & 50 \end{bmatrix}'$ m. The sensor collects $N = 100$ azimuth/elevation angle measurements, which ensure the target is observable. The initial sensor is located at $\mathbf{S}_1 = \begin{bmatrix} 0 & 0 & 0 \end{bmatrix}'$ m, the motion velocity components of the sensor in $x$, $y$ and $z$ directions are 0.5 m/s, 0.8 m/s, 1 m/s, respectively. The bearing angle measurement noise standard deviations $\sigma = \xi$ are set in between $\sqrt{0.01}$ and $\sqrt{0.08}$ radian with the variance difference of 0.01 radian$^2$. One thousand independent Monte Carlo experiments are run. We use the mean square error (MSE) and bias norm as evaluation indicators. The MSE and bias norm are defined as:

$$MSE = \frac{\sum_{i=1}^{1000} \sum_{j=1}^{3} (\hat{\mathbf{T}}_{\mathbf{i}}(j) - \mathbf{T}(j))^2}{1000}, \tag{90}$$

$$bias = \frac{\sum_{i=1}^{1000} \sum_{j=1}^{3} |\hat{\mathbf{T}}_{\mathbf{i}}(j) - \mathbf{T}(j)|}{1000}, \tag{91}$$

respectively. The first simulation is about the MSE and bias norm comparison between the proposed BC method and the original 3D localization method (PLE) in [15]. As shown in Figures 5 and 6, the BC method has lower MSE and bias norm compared to the PLE method. In addition, the MSE and bias norm increase as measurement noise level increases. The positioning accuracy of the two methods is significantly improved by using the weighted instrumental variable method. The PLE-WIV estimator, as expected, suffers from more serious bias than the BC-WIV estimator.

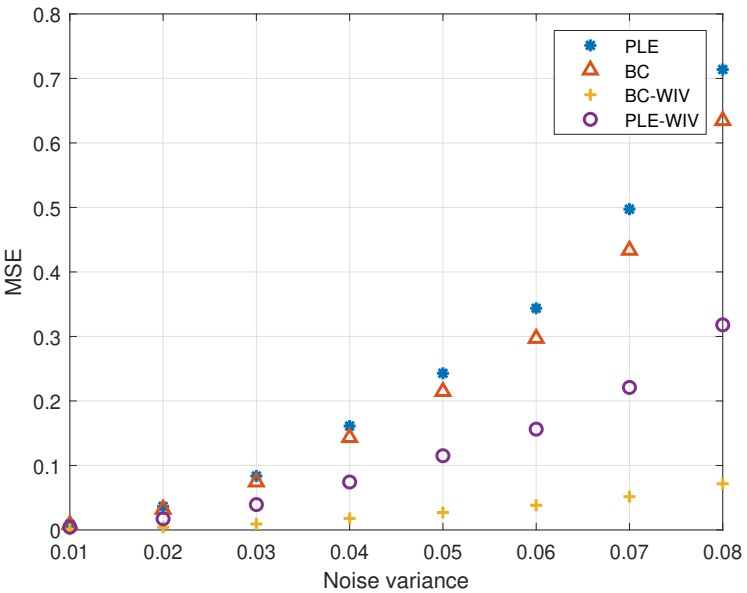

**Figure 5.** MSE with increasing noise standard deviation with a variance difference of 0.01 radian$^2$ for each method.

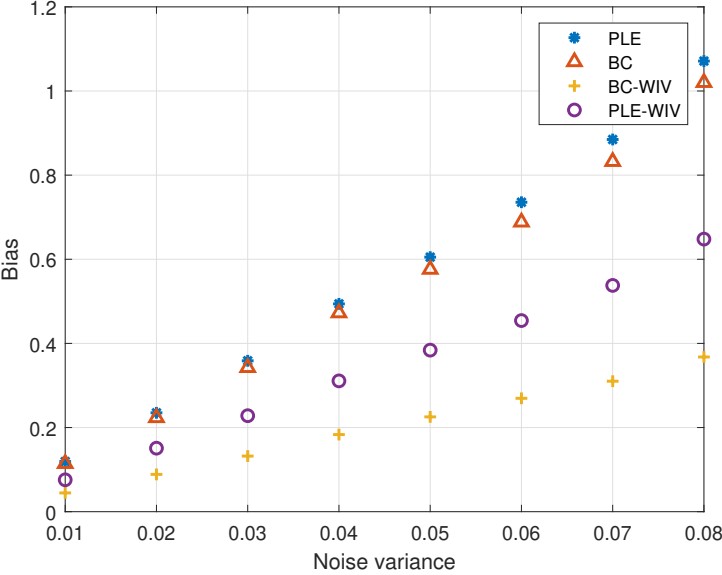

**Figure 6.** Bias norm MSE with increasing noise standard deviation with the variance difference of 0.01 radian$^2$ for each method.

The second simulation based on positioning methods is carried out with the increase of sensor number. The bearing angle measurement noise standard deviation is assumed to be $\sqrt{0.001}$ radian. The number of known sensors is changed from 60 to 260 with a difference of 40. The position of sensors is set as $x_{\mathbf{s}_i} \in (0 \ 50)$ m, $y_{\mathbf{s}_i} \in (0 \ 50)$ m, $z_{\mathbf{s}_i} \in (0 \ 50)$ m $(i = 1, \ldots, n)$.

Under the same conditions of each experiment, as shown in Figures 7 and 8, the BC and the BC-WIV methods still show the best positioning performance. The MSE and bias norm decrease when the number of sensors increases. However, there can be low positioning accuracy occasionally when the number of sensors is large. This is because in the process of the random generation of sensors, the position of some sensors makes the target unobservable, resulting in low positioning accuracy.

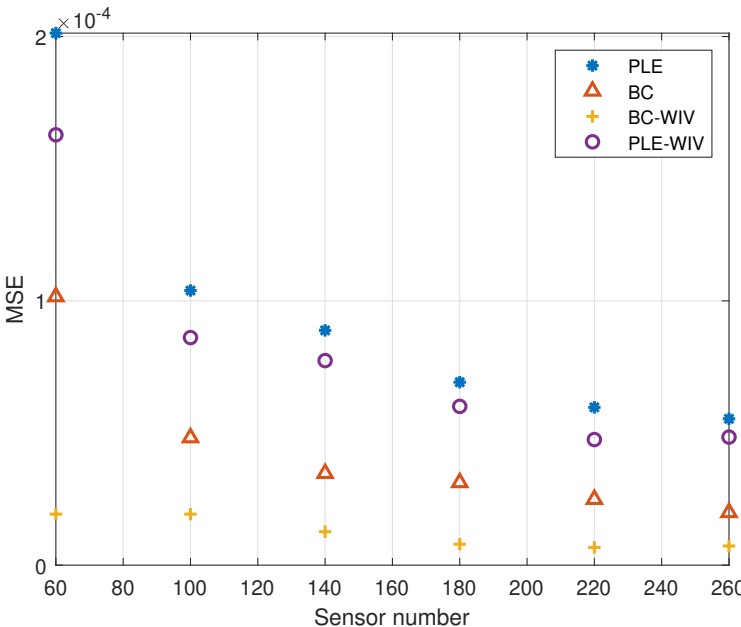

**Figure 7.** MSE with increasing sensor number at the difference of 40 for each method.

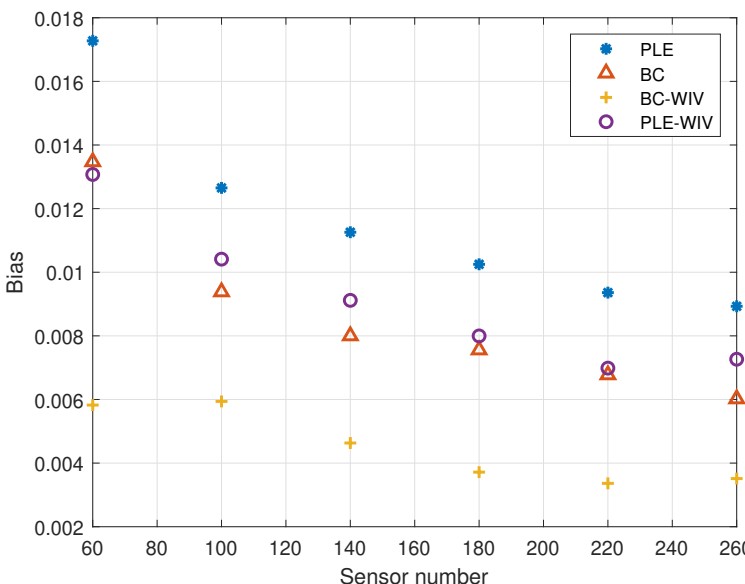

**Figure 8.** Bias norm with increasing sensor number at the difference of 40 for each method.

### 5.2. Example 2

The third simulation is about sensor trajectory planning. For fairness and completeness, we use the pseudolinear Kalman filter method in [30] for target localization as the platform to compare the performance of trajectories. One fitness function,

$$L(\hat{\mathbf{T}}) = 100 * (1 - \frac{||\mathbf{T} - \hat{\mathbf{T}}||_2}{||\mathbf{T}||_2}), \tag{92}$$

in [31] is used to demonstrate the performance of each estimator over $n$ runs. It could even be much less than zero when the estimators deviate far from the real values. To downplay the effect of the extremum points, $L(\hat{\mathbf{T}})$ is modified into:

$$\acute{L}(\hat{\mathbf{T}}) = \max(0, 100 * (1 - \frac{||\mathbf{T} - \hat{\mathbf{T}}||_2}{||\mathbf{T}||_2})). \tag{93}$$

**Trajectory 3.** *The location $\mathbf{S}_{k+1}$ on the trajectory is generated through the maximization of $det(\mathbf{FIM}_{k+1})$-$det(\mathbf{FIM}_{k-2})$ with respect to $\mathbf{S}_{k+1}$ recursively. The design of $\mathbf{S}_{k+1}$ is given by the $(k+1)$th measurement jointly with its two temporal nearest predecessors.*

**Trajectory 4.** *The location $\mathbf{S}_{k+1}$ on the trajectory is generated through the minimization of the state estimate covariance.*

The four trajectories planning methods are simulated and compared. The positions of the target and sensors are located at $\mathbf{T} = \begin{bmatrix} 20 & 35 & 55 \end{bmatrix}' \text{ m}$, $\mathbf{S}_1 = \begin{bmatrix} 10 & 25 & 20 \end{bmatrix}' \text{ m}$, $\mathbf{S}_2 = \begin{bmatrix} 30 & 20 & 40 \end{bmatrix}' \text{ m}$, $\mathbf{S}_3 = \begin{bmatrix} 70 & 50 & 15 \end{bmatrix}' \text{ m}$, $\mathbf{S}_4 = \begin{bmatrix} 12 & 80 & 70 \end{bmatrix}' \text{ m}$, $\mathbf{S}_5 = \begin{bmatrix} 59 & 22 & 18 \end{bmatrix}' \text{ m}$, $\mathbf{S}_6 = \begin{bmatrix} 120 & 150 & 90 \end{bmatrix}' \text{ m}$, respectively. For each test, the four trajectories share the same six initial sensor locations. The noise of the angle measurements are Gaussian independently distributed with zero mean and the standard deviation is $\sqrt{0.1}$ radian. The anti-collision distance and the maximum distance of sensor movement are set to $R = 5$ m, $r = 10$ m, respectively. The radius of the cylindrical non-flying area is set to $\rho = 0.5R$. One hundred independent Monte Carlo experiments are run.

Performance of the four trajectories is measured and displayed at two observation points when $k = 10$ and $k = 15$. Table 1 shows that Trajectory 1 and 2 contribute to greater $det(\mathbf{FIM}_{k+1})$ in over 90% of tests where trajectory 4 is used as a benchmark. This is consistent with the numerical results given in Table 2 where Trajectory 1 has the highest $\acute{L}(\hat{\mathbf{T}})$ at $k = 10$ and $k = 15$. Compared with Trajectory 3 and 4, the target estimations are more accurate under Trajectory 1 and 2 as the corresponding $\acute{L}(\hat{\mathbf{T}})$ values are closer to 100. The accuracy of target localization is related to $det(\mathbf{FIM}_k)$. The larger the $det(\mathbf{FIM}_k)$, the higher the positioning accuracy.

**Table 1.** Statistics that sum up the number of tests when $det(\mathbf{FIM}_k^i) - det(\mathbf{FIM}_k^4) > 0 (i = 1, 2, 3)$ for trajectory $i$ at two observations.

| Observation Point | $det(\mathbf{FIM}_k^1) -$ $det(\mathbf{FIM}_k^4) > 0$ | $det(\mathbf{FIM}_k^2) -$ $det(\mathbf{FIM}_k^4) > 0$ | $det(\mathbf{FIM}_k^3) -$ $det(\mathbf{FIM}_k^4) > 0$ |
|---|---|---|---|
| k = 10 | 98/100 | 96/100 | 3/100 |
| k = 15 | 99/100 | 96/100 | 2/100 |

**Table 2.** Mean values of $\acute{L}(\hat{\mathbf{T}})$ for four trajectories at two observation points.

| | Trajectory 1 | Trajectory 2 | Trajectory 3 | Trajectory 4 |
|---|---|---|---|---|
| mean of $\acute{L}(\hat{\mathbf{T}})(k = 10)$ | 91.16 | 90.58 | 89.62 | 90.49 |
| mean of $\acute{L}(\hat{\mathbf{T}})(k = 15)$ | 93.42 | 92.86 | 88.85 | 90.99 |

From Theorem 3, $det(\mathbf{FIM}_{k+1}) - det(\mathbf{FIM}_k)$ for trajectory 1 and $det(\mathbf{FIM}_{k+1}) - det(\mathbf{FIM}_{k-2})$ for Trajectory 3 both are parts of $det(\mathbf{FIM}_{k+1})$. Since Trajectory 3 only utilizes the sensor location of the past two moments, $det(\mathbf{FIM}_{k+1}) - det(\mathbf{FIM}_{k-2})$ weighs less in $det(\mathbf{FIM}_{k+1})$ than $det(\mathbf{FIM}_{k+1}) - det(\mathbf{FIM}_k)$. The maximizer of $det(\mathbf{FIM}_{k+1}) - det(\mathbf{FIM}_{k-2})$ is less likely to be close to the global optimizer of $det(\mathbf{FIM}_{k+1})$. Under such circumstances, Trajectory 3 contributes a smaller information increment than Trajectories 1 and 2. Trajectory 4 is achieved by minimizing the trace of the state estimation covariance matrix. Due to the inverse relation between the covariance and FIM, the generation of

Trajectory 4 also utilizes more past information than Trajectory 3. However, compared with Trajectories 1 and 2, not all the information is used.

Comparison of the mean of $\det(\mathbf{FIM_k})$ for four trajectories can be seen intuitively in Figure 9. When the sampling step increases, $\det(\mathbf{FIM_k})$ of the proposed method increases more. In addition, more $\det(\mathbf{FIM_k})$ can be obtained when the angles are updated in Trajectory 1 at the cost of increasing the calculation amount. Less computation time is consumed by generating Trajectory 2. When the measurement noise is small, the difference of the determinant obtained by Trajectories 1 and 2 is subtle as seen in the inner plot of Figure 9.

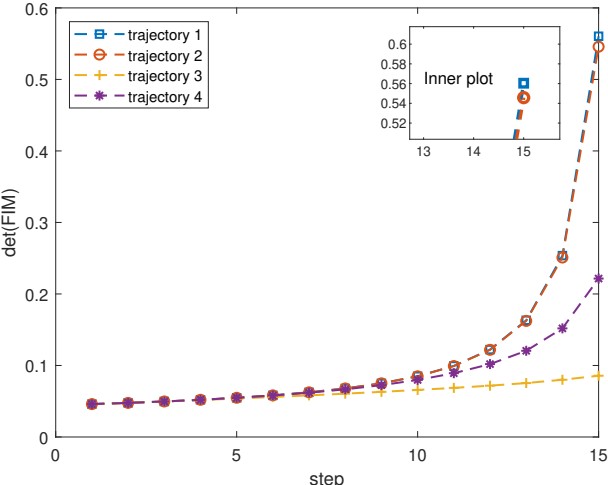

**Figure 9.** Comparison of the mean of $\det(\mathbf{FIM_k})$ for four trajectories over 100 Monte Carlo runs.

The trajectories simulation results are shown in Figure 10. In the process of trajectory planning, the predicted sensor position keeps getting closer to the target. When the sensor motion area is separated from the target safety area, $\mathbf{S}_{k+1}$ sits on the movement boundary sphere centering at $\mathbf{S}_k$.

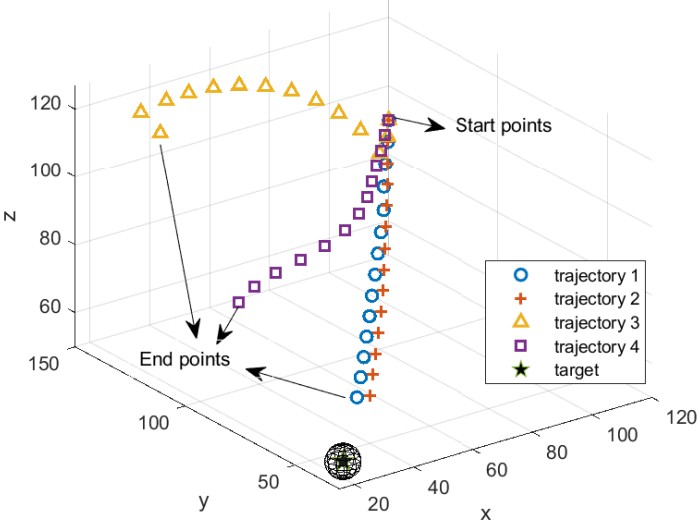

**Figure 10.** Sensor trajectories in typical runs. The trajectories in other runs are similar to the ones in this run. Trajectories 1, 2, 3 and 4 are represented by circle, cross, upper triangle and square, respectively. The black sphere is the boundary of the target safety area. The target position represented by the five pointed star.

As shown in Figure 10, Trajectories 1 and 2 approach the target much faster because the trajectory planning method proposed in this paper takes into account sensor position

information at all past moments and have higher FIM determinant. Since Trajectories 3 and 4 correspond to lower and mild FIM determinants, their curves reach the target in a slow and mild speed, respectively, amongst four trajectories; although it seems that Trajectory 3 distances itself from the target.

For each scenario, the whole computation complexity includes the following parts: initialization, Kalman-related localization, objective function construction and optimization. The average runtime for each trajectory is given in Table 3 to describe the complexity.

**Table 3.** Averaged runtimes for four trajectories.

|  | **Trajectory 1** | **Trajectory 2** | **Trajectory 3** | **Trajectory 4** |
|---|---|---|---|---|
| runtime(s) | 0.5558 | 0.5240 | 0.6996 | 0.5899 |

In Table 3, it can be seen that the time consumed for Trajectory 2 is the shortest. The generation of Trajectory 2 can effectively reduce the computational complexity in the optimization of (62) in return for no update of the bearing angles. The generation of Trajectory 3 takes a longer time since the objective function is different from Trajectories 1 and 2. For Trajectory 4, inverse operation of Fisher information matrix results in a longer calculation time.

## 6. Conclusions and Future Works

This paper studies the bearing-only localization from the improvement of target localization method and the reasonable planning of a sensor trajectory strategy. Under the condition of Gaussian noise, a potential relationship exists between the determinant of the extended coefficient matrix and the noise variance in the angle measurement equation. When the determinant of the extended coefficient matrix is zero, the variance of zero-mean Gaussian noise can be derived. Based on this estimated variance, the bias compensation localization of the target on the 3D space is realized. The estimator can be further refined by the BC-WIV method. In addition, a trajectory planning algorithm encapsulating two trajectory planning strategies are proposed to improve localization accuracy under several constraints. The next moment sensor position is given by optimizing quantified FIM determinant increment as the objective function that evaluates all the historical measurements. The optimal solution always lies on the constraint boundary. Simulation results show that Trajectory 1 leads to a higher localization accuracy at the cost of more computation time while Trajectory 2 has a comparable result with a mild computation burden.

In the future, we plan to extend our current approach to multiple sensors localization. The research work in this paper is only carried out at the theoretical level, the next research will test the method proposed in the practical communication environment. A team of three Pioneer robots is expected to be deployed in a rectangular area, with one Pioneer robot as the target and the other two Pioneers as tracking sensors to locate the target by the obtained bearing information.

## 7. Patents

Yiqun Zou, Bilu Gao, Xiafei Tang. 3D trajectory planning method, system, equipment and medium based on bearing-only measurements: 202111562286.9[P].2021.12.20.

**Author Contributions:** Conceptualization, Y.Z.; methodology, B.G.; software, B.G.; validation, Y.Z., B.G. and X.T.; formal analysis, Y.Z.; investigation, B.G.; resources, Y.Z.; data curation, Y.Z.; writing—original draft preparation, B.G.; writing—review and editing, B.G. and Y.Z.; visualization, X.T.; supervision, L.Y.; project administration, X.T.; funding acquisition, Y.Z. All authors have read and agreed to the published version of the manuscript.

**Funding:** This work is supported by National Natural Science Foundation of China (NSFC) [grant 61403427] and Hunan Provincial Natural Science Foundation of China [project 2020JJ5585 and project 2020JJ5777].

**Institutional Review Board Statement:** Not applicable.

**Informed Consent Statement:** Not applicable.

**Data Availability Statement:** Not applicable.

**Acknowledgments:** The authors are also grateful to Professor Erwei Bai in the university of Iowa for shedding light on some initial research in this paper.

**Conflicts of Interest:** The authors declare no conflict of interest. The funders had no role in the design of the study; in the collection, analyses, or interpretation of data; in the writing of the manuscript, or in the decision to publish the results.

## Abbreviations

The following abbreviations are used in this manuscript:

| | |
|---|---|
| FIM | Fisher information matrix |
| AOA | Angle of arrival |
| 2D | Two dimensional |
| 3D | Three dimensional |
| MSE | Mean square error |
| IV | Instrumental variable |
| CRLB | Cramér-Rao lower bound |
| UAV | Unmanned aerial vehicle |
| BC | Bias compensation |
| BC-WIV | Bias compensation weighted instrumental variable |
| PLE | Pseudo-linear estimator |

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
