# Peer review of "Target Localization and Sensor Movement Trajectory Planning with Bearing-Only Measurements in Three Dimensional Space"

_applsci, doi:10.3390/app12136739_

Round 1
Reviewer 1 Report
This paper deals with the AOA-based localization. However, it is unclear what the contribution of this paper is. The title, abstract, and introduction do not describe what problem to solve and why the problem is important. The English sentences in this paper are difficult to understand because of grammar errors. The authors should improve the English sentences with the help of a native English speaker or editing service. I think the manuscript is lengthy and recommend for the authors to delete unnecessary contents except the proposed algorithm.
Novelty and contribution of this paper are unclear. Lines 21-79 simply lists the results of existing research. Literature survey should specify the problem to be solved by the proposed algorithm and should describe the limitations of existing algorithms. The authors should explain what the problem (limitation) of existing algorithms. The abstract and introduction should be rewritten. This paper cannot be accepted unless the authors provide satisfactory revision.
Reviewer 2 Report
Dear Authors,
I found your article very interesting, but I suggest to introduce following remarks, which have to be added and fulfilled before publishing the paper:
1. Referring to citations in the text, I suggest to extend it a little bit not making mental shortcuts like “discussed in [9]”. It is better to mention the last name of article’s author.
2. What is the potential application of your approach of target localization? Can you provide examples, where can it be used?
3. Referring to previous remark, could you apply your approach on the real object and show the results of MSE?
4. In Conclusions I don’t see the further steps of your research.
5. There is one empty Chapter – “Patents”.
After improving above described issues in the paper I’d like to give my positive opinion on signing my review report.
Reviewer 3 Report
The authors study an interesting subject in literature: the problem of targeting localization, presenting a new method to compute the position in a 3D environment. However, some points can be improved:
It's interesting to discuss real applications that use similar methods to track sensor movement.
Taking into account the related work, what is the main difference between the proposed method and the others?
In the section results, the authors just point out the value differences among the methods. However, it's interesting to highlight the main features of each method explaining the curve behaviors.
In the simulations, what is the tool used to perform the replications? How many replications are computed?
Also, in the results in some parts, the authors discuss computation time without defining algorithm cost and present numerical results for it. " Less computation time is consumed by generating trajectory 2." How can we define computation time? What is the relevant operation to compute and determine the algorithm cost?
The paper's conclusion is summarized. Please, add more details about the methods' behavior and future works.
